# Taxol acts differently on different tubulin isotypes

Yean Ming Chew [1] & Robert A. Cross [1✉]

Taxol is a small molecule effector that allosterically locks tubulin into the microtubule lattice. We show here that taxol has different effects on different single-isotype microtubule lattices. Using in vitro reconstitution, we demonstrate that single-isotype α1β4 GDP-tubulin lattices are stabilised and expanded by 10 μM taxol, as reported by accelerated microtubule gliding in kinesin motility assays, whereas single-isotype α1β3 GDP-tubulin lattices are stabilised but not expanded. This isotype-specific action of taxol drives gliding of segmented-isotype GDP-taxol microtubules along convoluted, sinusoidal paths, because their expanded α1β4 segments try to glide faster than their compacted α1β3 segments. In GMPCPP, single-isotype α1β3 and α1β4 lattices both show accelerated gliding, indicating that both can in principle be driven to expand. We therefore propose that taxol-induced lattice expansion requires a higher taxol occupancy than taxol-induced stabilisation, and that higher taxol occupancies are accessible to α1β4 but not α1β3 single-isotype lattices.

[1] Centre for Mechanochemical Cell Biology, Warwick Medical School, Coventry CV4 7LA, UK. ✉email: r.a.cross@warwick.ac.uk

Microtubules are fundamental for eukaryotic life. They are essential for a wide variety of cellular functions, including chromosome segregation, directed vesicle and organelle transport, the establishment of cell polarity, ciliary beating, cell motility, and the patterning of early embryos. Microtubules self-assemble from GTP-tubulin heterodimers, which polymerise head-to-tail to form protofilaments and side-by-side to form a closed tube. Tubulation produces a stiff structure, allowing microtubules to define and support cell shape. The head-to-tail polymerisation of tubulin molecules gives microtubules an intrinsic polarity, whereby each microtubule has a fast-growing plus end that displays β tubulin and a slow-growing minus end that displays α tubulin (Fig. 1a). At the tips of growing microtubules, GTP-tubulin heterodimers are captured from solution and incorporated into the tip lattice. The GTP-tubulin tip lattice is structurally stable but converts continuously via GTP hydrolysis and Pi release into the GDP-tubulin core lattice, which is structurally unstable. This polymerisation-coupled conversion from GTP-tubulin to GDP-tubulin results in dynamic instability[1] (Fig. 1a), whereby stochastic breaches in the GTP-tubulin cap expose the underlying GDP-tubulin core[2], triggering a catastrophe and resultant rapid depolymerisation. Re-establishment of the stabilising GTP-tubulin cap (rescue) protects the unstable GDP-tubulin core and allows growth to resume. In cells, an array of microtubule end-binding proteins modulates dynamic instability[3].

Recent work has shown that in addition to classical dynamic instability, the processes of which occur at the microtubule tip, dynamic conformational changes in the core GDP-tubulin lattice of polymerised microtubules can also occur, driven by the binding of allosteric effectors[2,3]. Such conformational shifts cause expansion or compaction of the lattice, corresponding to changes in the average axial spacing between tubulin heterodimers. Aside from GTP, GDP.Pi, GDP[4,5] and their analogues (for example GMPCPP[6] and GTPγS[7]), allosteric effectors for tubulin conformation in the lattice include neighbouring tubulin molecules[8], a variety of lattice-binding macromolecules, for example EB proteins[9], doublecortin[9,10], kinesin molecular motors[11,12], tau[13], TPX2[14], CAMSAP3[15] and a number of lattice-binding drugs[16].

One such drug is taxol (generic name paclitaxel), the first microtubule stabilising drug to be discovered[17]. Taxol remains a critically-important chemical biological tool for the chemotherapy of breast, lung and ovarian cancer, and for basic science. In mixed-isotype brain microtubules, taxol not only stabilises the core GDP-tubulin lattice, but also expands the axial spacing between tubulin heterodimers compared to that of the unliganded GDP-tubulin lattice[18]. Taxol binds β tubulin, on the inside (luminal) surface of microtubules, adjacent to the M-loop, which connects laterally between protofilaments. Taxol shares its binding site with other taxanes such as docetaxel (Taxotere®) and cabazitaxel (Jevtana®) and with non-taxanes such as epothilone. Taxol is proposed directly to stabilise lateral connections between protofilaments and allosterically to stabilise the longitudinal interface between neighbouring heterodimers in a protofilament[5,19–21]. Crystallographically precise details of the taxol binding site are now established[22], but the molecular mechanisms by which taxol shifts the conformation of lattices of polymerised tubulin heterodimers are less clear. Exactly how taxol works in cells is also not fully clear. Taxol was initially thought to arrest mitosis by activating the spindle assembly checkpoint, ultimately resulting in cell death. More recently, taxol has been shown to cause multipolar divisions, and thereby aneuploidy and the consequent death of daughter cells with incomplete sets of chromosomes[23].

There are indications that the extent or magnitude of taxol-induced lattice expansion can depend on whether taxol binds before or after the lattice assembles[24]. Ordinarily, microtubule assembly requires GTP, but taxol can drive the assembly of GDP-tubulin into microtubules[25,26], and can stabilise short protofilaments in solution[27]. Fluorescent analogues of taxol have been shown to create and/or bind preferentially to lattice defects[28]. Defects have their own, local dynamics, allowing new tubulin to be incorporated in a repair process[28].

Humans have 9 α and 10 β tubulin isotypes[29,30]. It is unclear why humans need so many tubulins, but we do: single point mutations in specific human tubulin isotypes are documented to cause specific human diseases (tubulinopathies)[31], often linked to developmental abnormalities[32,33]. Taxol resistance in patients can be accompanied by an increase in β3 tubulin expression[34], suggesting that β3 tubulin expression can protect against taxol[26,35]. There is firm evidence that β3 tubulin within mixed-isotype microtubules binds less taxol than other isotypes in the lattice[36]. Here we show that single-isotype human α1β3, zebrafish α1β4 and human α1β4 microtubules have different lattice stabilities and different lattice-mechanical responses to taxol. By studying single isotype microtubules, mosaic-isotype microtubules and segmented-isotype microtubules, we probe the spatial range of taxol-dependent conformational switching in the different lattices, with relevance to the molecular mechanism of taxol-induced microtubule stabilisation, to the different biological functions of different tubulin isotypes, and to cancer medicine.

## Results

We have sought here to compare α1β3 and α1β4 metazoan tubulin isotypes. β3 tubulin is confined largely to neurons[35] and is of interest for this reason and by virtue of its potential role in taxol resistance. β4b tubulin has a broader tissue specificity[37]. We worked with both human (*Homo sapiens*, Hs) and zebrafish (*Danio rerio*, Dr) β4b isotypes, to test whether their very minor sequence differences influence function. Tubulin mutants in *Danio rerio* can potentially illuminate human tubulinopathies[29,32]. Hereafter, "Dr α1β4" indicates zebrafish α1cβ4b and "Hs α1β4" indicates human α1bβ4b, whilst "α1β4" indicates both. "Hs α1β3" indicates human α1bβ3. To find out if the sequence differences between Hs α1β3 and Dr/Hs α1β4 tubulins affect the molecular action of taxol, we assembled defined-isotype lattices and compared their in vitro properties and the influence of taxol on those properties. We expressed tubulins in insect cells and used dual affinity tag-purification, with an 8x his tag and a FLAG tag on the C-termini of α and β tubulins, respectively (see "Methods" and ref. [38]), to ensure we obtained only defined-isotype full-length tubulin heterodimers. Experiments quantifying dynamic instability used dark field illumination of unlabelled tubulins. For kinesin-driven microtubule gliding assays we used epifluorescence imaging, adding 5% HiLyte 488 or HiLyte 647porcine brain tubulin as a trace fluorescent label (see "Methods"). Single isotype tubulins were not fluorescently labelled.

**α1β3 and α1β4 single isotype microtubules differ in their dynamic instability.** α1β3 and α1β4 single isotype microtubules nucleated from porcine brain GMPCPP-microtubule seeds (Fig. 1b) have markedly different dynamics. α1β3 microtubules depolymerise faster than α1β4 microtubules (Fig. 1c, d). In both α1β3 and α1β4 microtubules, plus end depolymerisation following catastrophe is usually monophasic, but α1β4 microtubules can depolymerise in multiple phases (Fig. 1e), implying a structural difference between the corresponding regions of the lattice, for example a change in protofilament number, occurring at a lattice defect[39]. Switching of depolymerising microtubules to a slower depolymerisation rate was recently described[40]. Of note, for in vitro assembled single-isotype Dr α1β4 microtubules, we see examples of switching to faster depolymerisation, as well as switching to slower depolymerisation (Fig. 1e). Growth rates of Hs α1β3 and Dr α1β4 microtubules differ only by ~15%, albeit the difference is significant

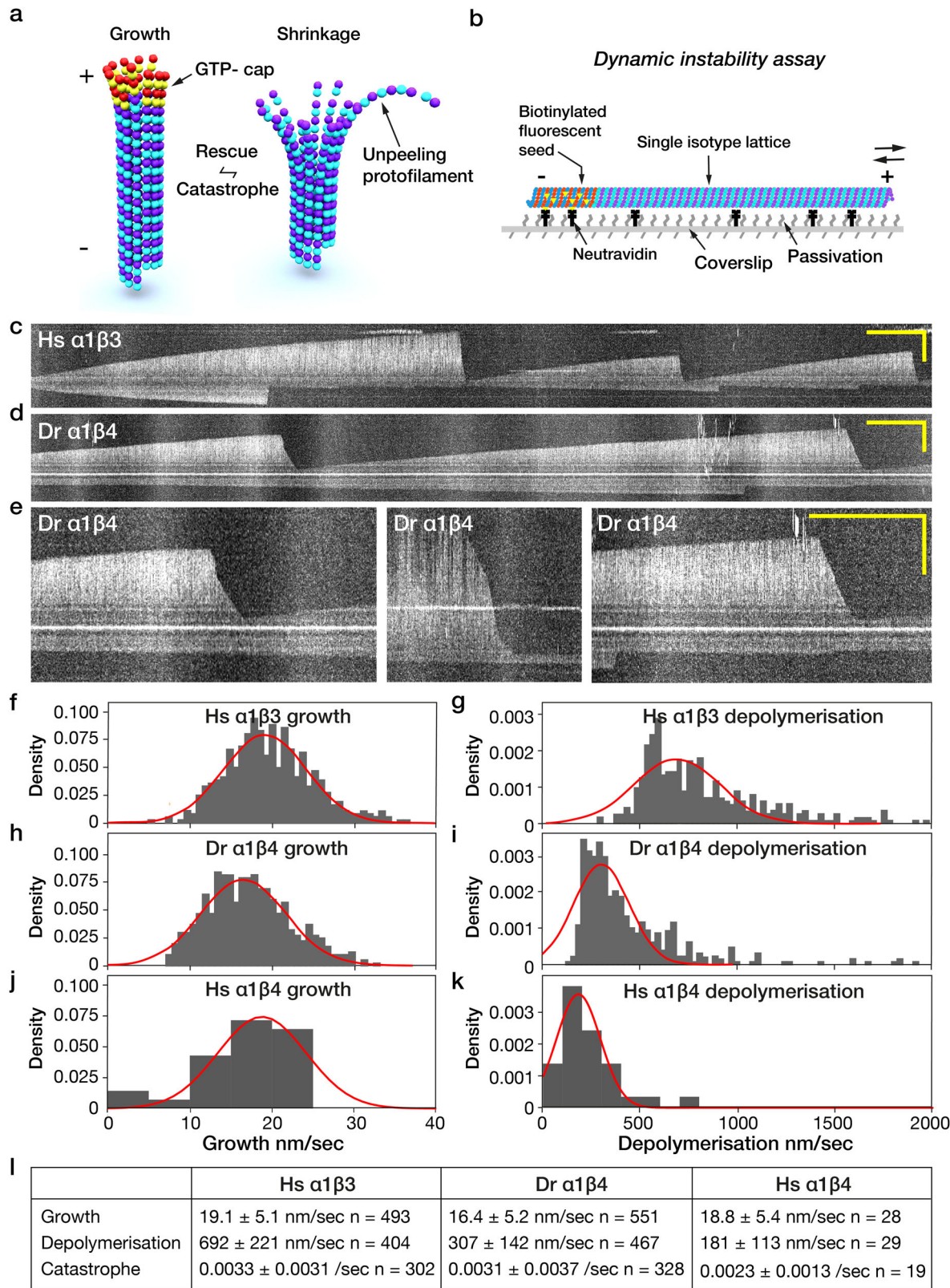

| | Hs α1β3 | Dr α1β4 | Hs α1β4 |
|---|---|---|---|
| Growth | 19.1 ± 5.1 nm/sec n = 493 | 16.4 ± 5.2 nm/sec n = 551 | 18.8 ± 5.4 nm/sec n = 28 |
| Depolymerisation | 692 ± 221 nm/sec n = 404 | 307 ± 142 nm/sec n = 467 | 181 ± 113 nm/sec n = 29 |
| Catastrophe | 0.0033 ± 0.0031 /sec n = 302 | 0.0031 ± 0.0037 /sec n = 328 | 0.0023 ± 0.0013 /sec n = 19 |

(Fig. 1f, h). Depolymerisation rates are markedly different: α1β3 microtubules (Fig. 1g) depolymerise much faster than Dr α1β4 microtubules (Fig. 1i) or Hs α1β4 microtubules (Fig. 1k). Note that we fit both growth and depolymerisation histograms with gaussians, but for depolymerisation the fits are poor, consistent with multiple phases of depolymerisation (including, for Hs α1β3 lattices, multiple types of monophasic event). Figure 1l tabulates

values for the parameters of dynamic instability for these single-isotype microtubules.

**Less taxol is required to stabilise α1β4 GDP-lattices than α1β3 GDP-lattices**. We find, as previously suggested[34] that Hs α1β3 lattices are relatively insensitive to taxol. 50 nM taxol has no

**Fig. 1 α1β3 and α1β4 single isotype microtubules differ in their dynamic instability. a** Schematic of dynamic instability. Microtubules grow by addition of GTP-tubulin to their ends, forming a stabilising cap (red & yellow dimers). Loss of this cap (catastrophe) exposes the unstable GDP-tubulin core of the microtubule, which shrinks via rapid loss of curved GDP-tubulin subunits. Re-establishment of the cap (rescue) reverts the microtubule to steady growth. **b** Schematic of flow cell assay of microtubule plus end dynamics. **c–e** Kymographs showing spontaneous catastrophe and depolymerisation of **c** Hs α1β3 and **d, e** Dr α1β4 single-isotype microtubules (white). **e** Examples of 2-phase, 3-phase and 1-phase depolymerisation of Dr α1β4 lattices. Leftmost and rightmost views are zooms from **d**. Scale bars 10 μm (vertical) and 2 min (horizontal). **f, h, j** Growth rates for **f** Hs α1β3, **h** Dr α1β4 and **j** Hs α1β4 microtubules at 12 μM tubulin. **g, i, k** Depolymerisation rates for **g** Hs α1β3, **i** Dr α1β4 and **k** Hs α1β4 are clearly different. **l** Plus end dynamic instability parameters of α1β3 and α1β4 single-isotype microtubules. Rates are given as mean ± SD. Hs α1β3 and Dr α1β4 data from 3 experiments, Hs α1β4 from 2 experiments.

detectable effect on dynamically unstable Hs α1β3 microtubules (Fig. 2a). By contrast, 50 nM taxol is highly effective at stabilising Dr α1β4 microtubules against depolymerisation, abrogating catastrophe and driving continuous (processive) growth (Fig. 2b). Taxol thus amplifies an intrinsic difference in the stability of Dr α1β4 and Hs α1β3 microtubule lattices. For Hs α1β3 microtubules, 250 nM taxol partially inhibits catastrophe, slows depolymerisation and promotes rescue (Fig. 2c, yellow arrows). 500 nM taxol fully inhibits Hs α1β3 catastrophe and significantly reduces the rate of plus end growth from 19.4 ± 5.3 nm/s (median ± SD, $n = 20$) at 12 μM tubulin in the absence of taxol to 15.6 ± 7.2 nm/s (median ± SD, $n = 20$) in its presence (Fig. 2d). For minus ends in the absence of taxol (Fig. 2d), a population is present for which growth is too slight to measure and is assigned as zero. This population disappears on adding taxol, reflecting that 500 nM taxol converts both ends of Hs α1β3 microtubules to processive growth. Growth rates of Dr α1β4 microtubules were not obtained because the lattice develops kinks at this taxol concentration (below).

**Taxol differentially stabilises the segments of β3/β4 segmented-isotype microtubules.** To probe for axial communication between adjacent single-isotype regions of the same microtubule lattice, we assembled microtubules made of two contiguous single-isotype segments, a proximal α1β3 tubulin segment and a distal Dr α1β4 segment (Fig. 2e). At 100 nM taxol, α1β3 lattices remain susceptible to catastrophe, but the incidence of catastrophe is reduced, which allows us to obtain enough initial α1β3 segments of sufficient length. α1β3 tubulin is then washed out with Dr α1β4 tubulin and the second segment assembled, whilst maintaining taxol at 100 nM. On washout of free tubulin, the single-isotype segments depolymerise sequentially in clearly distinct phases, with the proximal Hs α1β3 segments depolymerising faster than the distal Dr α1β4 segments (Fig. 2f). The transition between phases occurs abruptly, with no detectable tendency of the proximal, intrinsically faster-depolymerising Hs α1β3 segments to accelerate the depolymerisation of the distal Dr α1β4 segment close to their intersection. Taxol differentially inhibits depolymerisation of the two segments, broadly in line with its effect on the corresponding single-isotype microtubules (Fig. 2g). In 100 nM taxol, proximal α1β3 segments depolymerise ~8-fold faster than distal Dr α1β4 segments (Fig. 2h).

**500 nM taxol puts kinks into α1β4 microtubules; taxol washout relaxes the kinks.** When assembled in 500 nM taxol, Dr α1β4 microtubules acquire kinks, whereas Hs α1β3 microtubules do not (Fig. 3a, b and Supplementary Movies 1 and 2). Even when assembled at 5 μM taxol, kinks were not observed in Hs α1β3 microtubules. At 100 nM taxol, kinks were absent from Dr α1β4 microtubules, suggesting that taxol concentration must exceed a threshold to cause kinking of microtubules. Kinky Dr α1β4 microtubules assembled in 500 nM taxol straighten upon washing out taxol using a solution of free tubulin only, showing that

kinking is reversible (Fig. 3c and Supplementary Movie 3). Washout with 100 nM taxol in the absence of free tubulin also relaxes kinks. We hypothesise that kinks are produced by localised, taxol-induced expansion of the Dr α1β4 lattice, which reverses upon taxol washout (Fig. 3d).

**Taxol accelerates the gliding motility of α1β4 but not α1β3 microtubules.** Kinesin-1 (Khc) can sense a taxol-dependent difference in the conformation of α1β3 versus α1β4 GDP-microtubules, which emerges as different microtubule gliding rates. At 200 nM taxol, both Hs and Dr α1β4 GDP-microtubules glide over a surface of full-length kinesin-1 dimers at ~450 nm/s (Fig. 4a). Quantification of gliding at 200 nM taxol is less precise because microtubules depolymerise as they glide. Gliding velocity is perhaps 5–10% underestimated due to this effect. In 10 μM taxol, both Hs α1β4 and Dr α1β4 GDP-microtubules accelerate to almost 800 nm/s, whereas Hs α1β3 microtubules do not (Fig. 4a). We could not measure the gliding velocity of Hs α1β3 microtubules at submicromolar taxol concentrations as the microtubules depolymerised too rapidly. For α1β4 microtubules, individual microtubules at any taxol concentration tend strongly to move at either the fast or the slow speed (Fig. 4a). GMPCPP microtubules, both Hs α1β3 and Dr α1β4, all move at the faster rate (Fig. 4a).

**Gliding segmented microtubules "beat" when the α1β3 segment leads.** Given that taxol-stabilised Hs α1β3 and Dr α1β4 GDP-microtubules glide at different speeds on kinesin-1 (Khc) surfaces, we asked what happens to segmented-isotype microtubules gliding on the same surfaces (see "Methods"). We find that with the slower Hs α1β3 segment leading, the intrinsically faster Dr α1β4 segment tends to deviate sideways into loops (Fig. 4b and Supplementary Movie 4). Addition of ADP makes the loops formed by the trailing Dr α1β4 segments larger (Fig. 4c and Supplementary Movie 4), potentially reflecting disengagement of some of the kinesins. With a Hs α1β3 segment leading a porcine brain (PB) segment, similar behaviour is seen (Fig. 4d, e and Supplementary Movie 5). We hypothesise that the meandering of the trailing Dr α1β4 segments is driven by the need to balance the net forwards force from the intrinsically faster trailing segment with the net resistive force due to the leading Hs α1β3 segment. With an Hs α1β3 segment leading, segmented PB-Hs α1β3 microtubules glide significantly faster than single-isotype Hs α1β3 microtubules, consistent with their Hs α1β3 leading segments being pushed from behind (Fig. 4f). Conversely, with a PB segment leading an Hs α1β3 segment, gliding is slower than for PB microtubules alone, consistent with drag from the trailing Hs α1β3 segment. We used segmented Hs α1β3-PB MTs microtubules for these experiments because they formed more readily than Hs α1β3-Dr α1β4 segmented microtubules.

**Mosaic-isotype microtubules glide more sinuously than single-isotype microtubules.** In the presence of 10 μM taxol, gliding

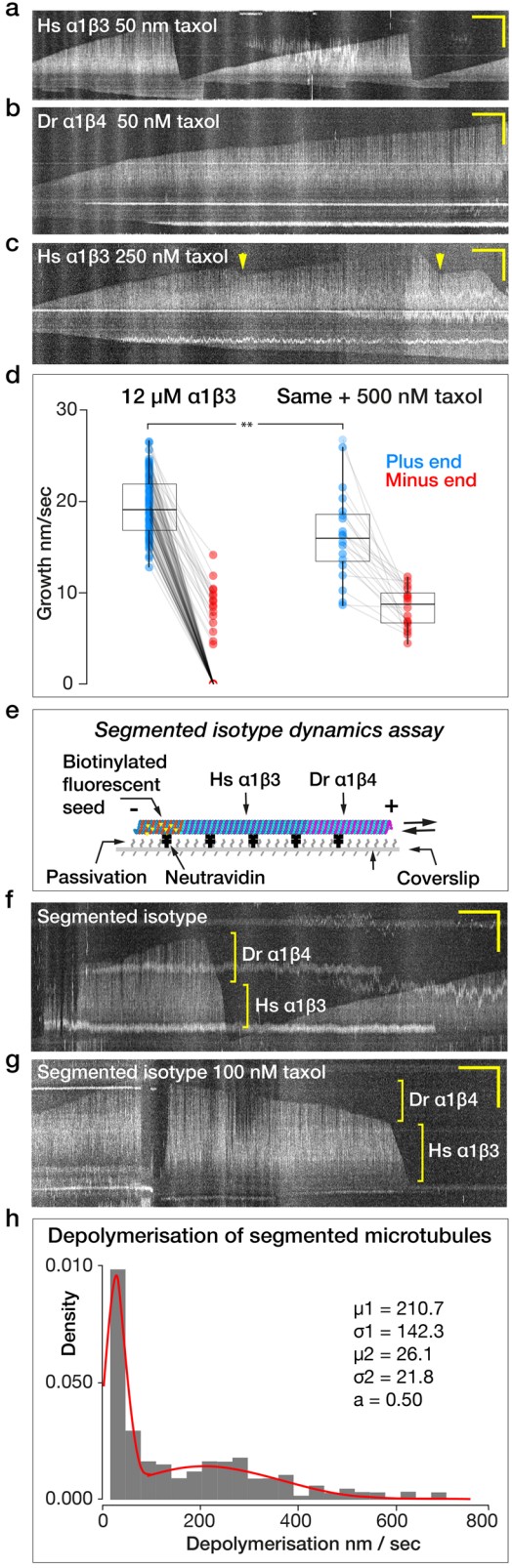

**Fig. 2 α1β3 and α1β4 microtubule lattices respond differently to taxol.**
Kymographs showing **a** dynamic instability of a Hs α1β3 microtubule in 50 nM taxol, **b** dynamic instability of a Dr α1β4 microtubule in 50 nM taxol. **c** Dynamic instability of a Hs α1β3 microtubule in 250 nM taxol. Catastrophes still occur but depolymerisation is slower, and rescues (yellow arrows) are seen. **d** Plus end (blue) and minus end (red) growth rates of Hs α1β3 microtubules at 12 μM tubulin, in the absence of taxol versus in the presence of 500 nM taxol. Each data point represents the mean growth rate during the growth phase of an individual microtubule. Growth rates with and without taxol are similar but significantly different. **e** Schematic of segmented-isotype microtubule dynamics assay. **f**, **g** Biphasic depolymerisation of segmented-isotype microtubules **f** in the absence of taxol and **g** in 100 nM taxol. Scale bars for all kymographs 2 min (horizontal) and 10 μm (vertical). **h** Depolymerisation rates of segmented-isotype microtubules in 100 nM taxol. μ1,2: mean depolymerisation rates (nm/s); σ: standard deviation; a: area under the curve of one population. Data in (**d**) and (**h**) obtained from 3 experiments. ns indicates $p \geq 0.05$ and significance is **$p < 0.01$. For box plot, boxes boxes mark interquartile range, whiskers mark SD, horizontal bar marks population median. All the data are plotted.

**Depolymerisation rates of mosaic-isotype microtubules depend nonlinearly on tubulin isotype ratio.** Mosaic microtubules obtained by copolymerising Hs α1β3 and Dr α1β4 tubulins blend the properties of the two isotypes, consistent with previous observations on mixing α1bβ1 + α1bβ4 and α1aβ3 human tubulin isoforms[41]. Strikingly, depolymerisation rates depend nonlinearly on the Hs α1β3:Dr α1β4 isotype ratio in the assembly mix (Fig. 5a). Growth rates were similar but significantly different for 100% Hs α1β3 and 100% Dr α1β4 microtubules (19.4 ± 5.3 and 16.8 ± 5.0 nm/s, respectively, at 12 μM tubulin, median ± SD) (Fig. 5b; see Fig. 1f for histograms).

**Gliding velocity of taxol-stabilised mosaic-isotype microtubules depends nonlinearly on tubulin isotype ratio.** In mosaic-isotype Hs α1β3:Dr α1β4 microtubules at 10 μM taxol, the speed of kinesin-driven microtubule gliding also depends nonlinearly on the isotype ratio in the assembly mix (Fig. 5d).

## Discussion

We find that taxol inhibits the depolymerisation of both α1β3 and α1β4 lattices, but α1β3 lattices require around tenfold more taxol than α1β4 entirely to suppress catastrophe. Further, taxol substantially accelerates the gliding speed of α1β4 microtubule lattices on surfaces of processive kinesin-1 dimers, but does not accelerate α1β3 lattices. What are the molecular origins of these different responses of α1β3 and α1β4 lattices, and what are their implications?

The differing responses of α1β3 and α1β4 microtubules to taxol show that a small set of conservative residue substitutions (Fig. 6a) shifts the apparent affinity of microtubules for taxol, as reported by the taxol-dependence of their gliding velocity. Away from the C-terminal tail sequences, most of these residue substitutions are surface-exposed and clustered in the regions of β tubulin that engage the M-loop of the lateral nearest neighbour in the lattice (Fig. 6b, c and Supplementary Movie 7). Crucially, the residue substitutions that distinguish β3 from β4 tubulin seemingly disfavour but do not fully inhibit taxol binding to the α1β3 lattice. Taxol at sufficient concentration does stabilise α1β3 microtubules, showing that taxol does bind. But even at 10 μM taxol, close to the limit of taxol solubility, kinesin-driven gliding of the α1β3 lattice is not accelerated.

The clearly different actions of taxol on α1β3 and α1β4 microtubules contrast with the action of GMPCPP, which both

mosaic microtubules built from a 50:50 mix of Hs α1β3:Dr α1β4 isotypes appear more sinuous than single-isotype microtubules of either type (Fig. 4g–i and Supplementary Movie 6). Quantification indicates the differences are significant (Fig. 4j). This might indicate a reduction in bending stiffness, or it might reflect a different pattern of kinesin-generated force.

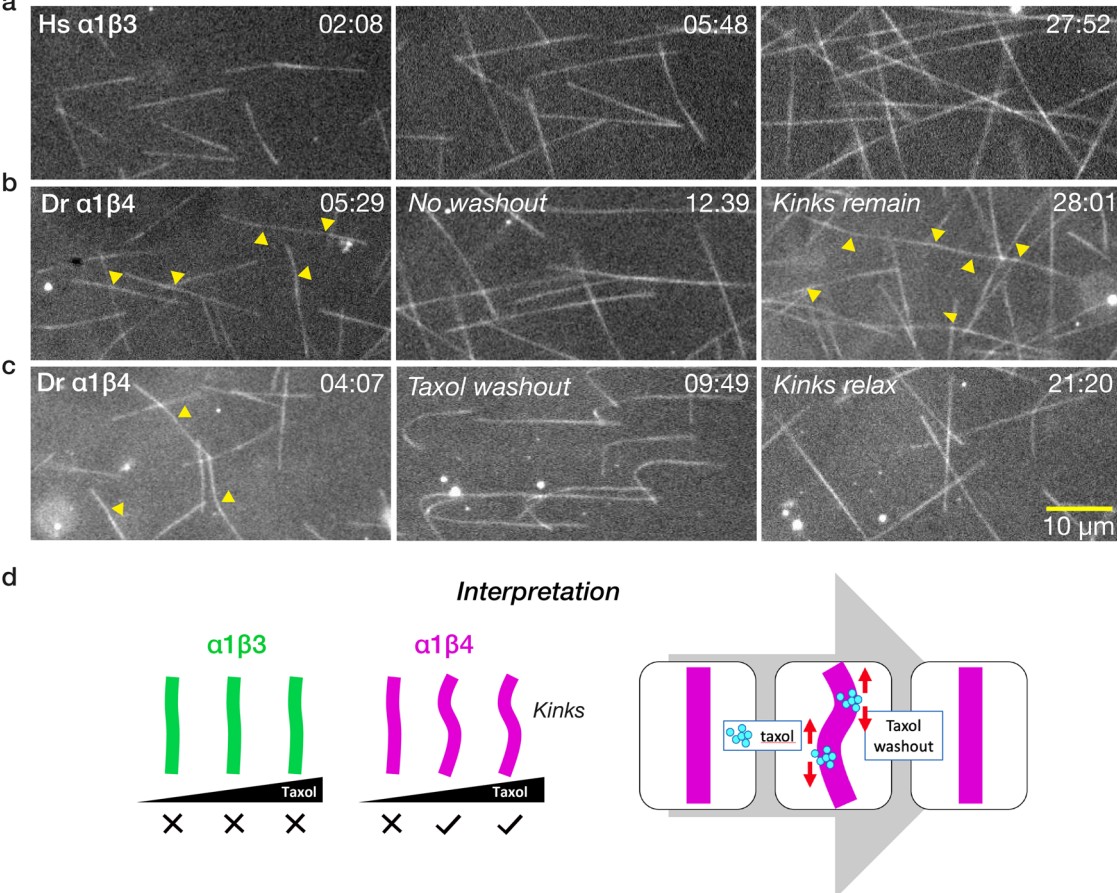

**Fig. 3 Taxol puts kinks into α1β4 but not α1β3 microtubules. a** Hs α1β3 microtubules grow processively and straight in the presence of 500 nM taxol at 12 μM tubulin (Supplementary Movie 1). **b** Under the same conditions, Dr α1β4 microtubules develop kinks (yellow arrows, Supplementary Movie 2). **c** Washing out taxol with free tubulin relaxes the kinks (Supplementary Movie 3). Scale bar 10 μm. Times in min:sec. **d** Schematic model. Local binding of taxol, possibly at defects, produces kinks by locally expanding the lattice.

stabilises the lattice and accelerates its kinesin-driven microtubule gliding, for both α1β3 and α1β4 microtubules (Fig. 4a). This demonstrates that with GMPCPP as the effector, both α1β3 and α1β4 lattices can shift into an expanded, faster-gliding conformation. We note that in contrast to GMPCPP, GMPPCP does not expand the lattice of mixed-isotype brain microtubules[42].

Together, these findings suggest a working model in which taxol binding stabilises both α1β3 and α1β4 lattices but shifts only α1β4 lattices into a faster-gliding conformation (Fig. 6d). In this picture, the GDP-tubulin lattice can access two effector-stabilised conformational states, with the favoured state dependent on the effector, the effector concentration, and the tubulin isotype. We postulate that only α1β4 lattices, and not α1β3 lattices, have a high enough taxol affinity to be shifted by achievable concentrations of taxol. Exactly how switching between lattice states is driven by taxol binding to individual heterodimers within the lattice remains to be seen. In mixed-isotype brain microtubules, maximum expansion requires stoichiometric amounts of taxol[22]. Our finding that sub-stoichiometric taxol can induce kinks in single-isotype α1β4 microtubules suggests that taxol can induce local expansion of an otherwise compacted single-isotype lattice. Our working model does not address the mechanism(s) by which the lattice can be driven to change conformation—it seeks only to encapsulate a minimal hypothesis in which the GDP-tubulin lattice can be shifted between exactly 2 different effector-stabilised conformations. It is possible that more than two effector-stabilised states of the GDP-tubulin lattice exist; but we only need two to explain our data.

The inhibition of α1β3 microtubule plus end growth by taxol (Fig. 2d) is consistent with previous reports[43]. The mechanism of this effect will require further investigation.

In this work we have not directly measured lattice spacings. Nonetheless, we hypothesise that the effector-dependent transition between two stabilised GDP-lattice conformations in our model corresponds to the much-studied transition from a compacted to an expanded microtubule lattice. We note that "expanded" and "compacted" are catch-all descriptors for complex conformational changes that include lateral as well as axial components. There is robust evidence that both taxol and GMPCPP expand the lattice of mixed-isotype brain microtubules[18], and that GMPCPP brain microtubules glide faster than GDP brain microtubules[44] over kinesin-1 surfaces. Here, we saw that taxol induces kinks in Dr α1β4 but not Hs α1β3 microtubules and that taxol washout relaxes the kinks (Fig. 3), consistent with them being due to reversible, taxol-dependent local expansion of the α1β4 single-isotype GDP-tubulin lattice, possibly at defects. We have therefore named our 2 taxol-stabilised GDP-lattice states, SLOW-gliding/Compacted and FAST-gliding/Expanded (Fig. 6). In our model, taxol-dependent stabilisation and taxol-dependent acceleration/expansion are separable. This is evident for both our isotypes. α1β4 lattices are stabilised but not accelerated at low taxol concentrations and accelerated at higher (10 μM) taxol concentrations. α1β3 lattices are stabilised by 10 μM taxol, but not expanded/accelerated. It is already known that effector-dependent stabilisation and effector-driven expansion of the microtubule lattice are not necessarily

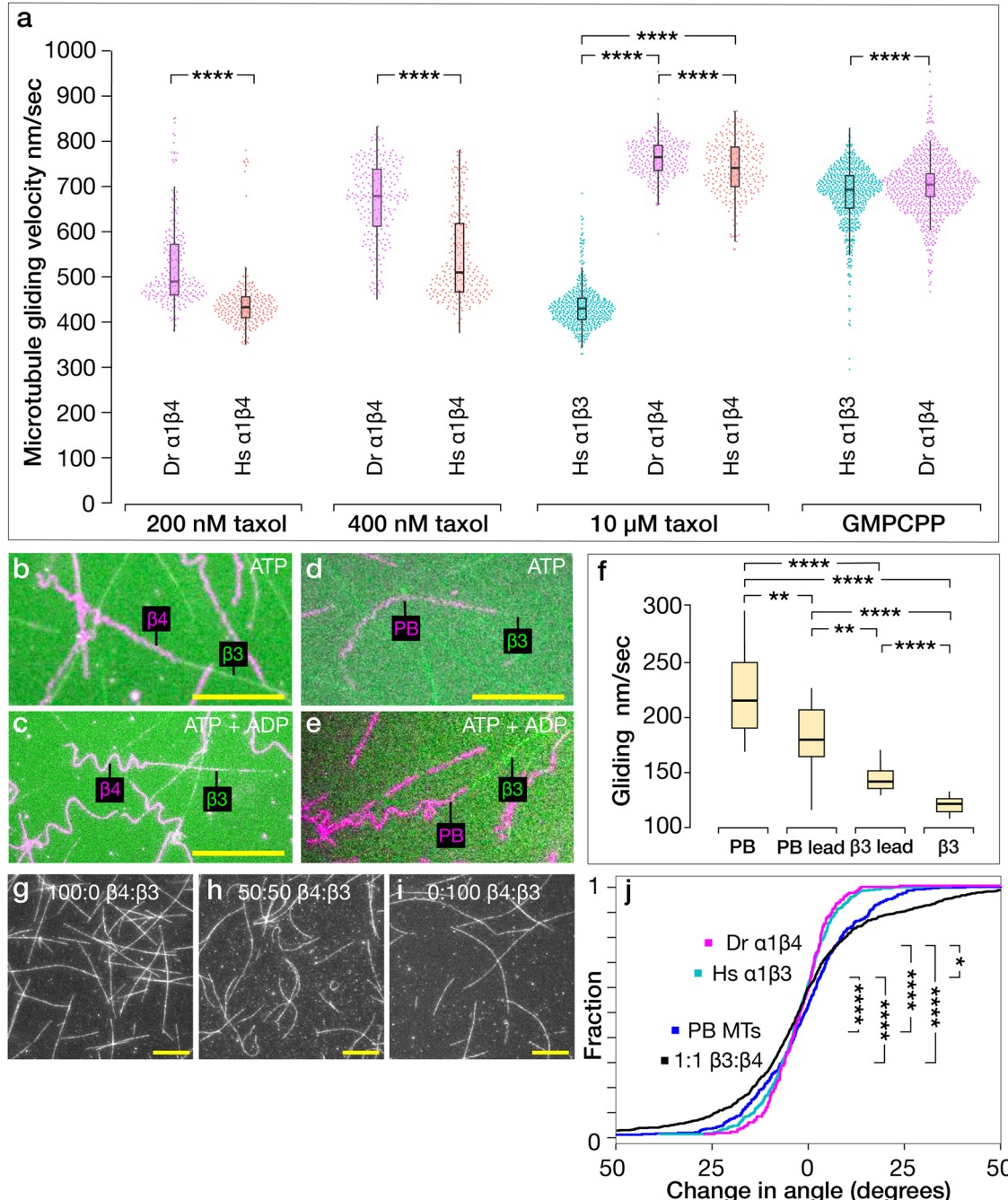

**Fig. 4 Taxol accelerates kinesin-driven gliding for α1β4 but not α1β3 lattices. a** Higher taxol concentrations accelerate α1β4 microtubules, both Hs and Dr. At 200 nM taxol, Dr α1β4 microtubules glide at 491 ± 96 ($n = 231$) nm/s; Hs α1β4 microtubules glide at 434 ± 65 ($n = 238$) nm/s. At 400 nM taxol, Dr α1β4 microtubules glide at 680 ± 87 ($n = 219$) nm/s; Hs α1β4 microtubules glide at 511 ± 99 ($n = 229$) nm/s. At 10 μM taxol, Hs α1β3 microtubules glide at 431 ± 49 ($n = 505$) nm/s, Dr α1β4 microtubules glide at 766 ± 42 ($n = 242$) nm/s; Hs α1β4 microtubules glide at 742 ± 63 ($n = 236$) nm/s. GMPCPP Hs α1β3 and Dr α1β4 microtubules both slide at the faster rate at 694 ± 69 ($n = 653$) and 705 ± 44 ($n = 690$) nm/s, respectively. $n$ is the number of instantaneous velocity values, see "Methods". Quoted velocities are median ± SD. Data collected from 3 experiments, except for Hs α1β3 microtubules which were collected from 2 experiments. **b–e** Segmented-isotype gliding assays. Trailing Dr α1β4 segments (**b**, **c**, Supplementary Movie 4) or trailing porcine brain (**d**, **e**, Supplementary Movie 5) segments of gliding segmented-isotype microtubules meander behind their leading Hs α1β3 segments. Scale bars 20 μm. **f** Gliding velocities of segmented porcine brain (PB) - Hs α1β3 microtubules, in 2 mM ATP plus 2 mM ADP: PB segment leading 180 ± 32 ($n = 18$), PB segment only 216 ± 42 ($n = 20$), α1β3 segment leading 142 ± 15 ($n = 15$), α1β3 segment only 122 ± 8 ($n = 16$) nm/s. Velocities are median ± SD. **g–i** Gliding of mosaic-isotype microtubules (Supplementary Movie 6). **j** Empirical cumulative density functions comparing the change in gliding direction of single-isotype versus mosaic-isotype microtubules (see "Methods"). Mixed-isotype lattices appear significantly more sinuous. ns indicates $p \geq 0.05$ and significance is *$p < 0.05$, **$p < 0.01$, and ****$p < 0.0001$. In box plots, boxes indicate interquartile range, whiskers mark SD, horizontal bar marks population median. All the data are plotted.

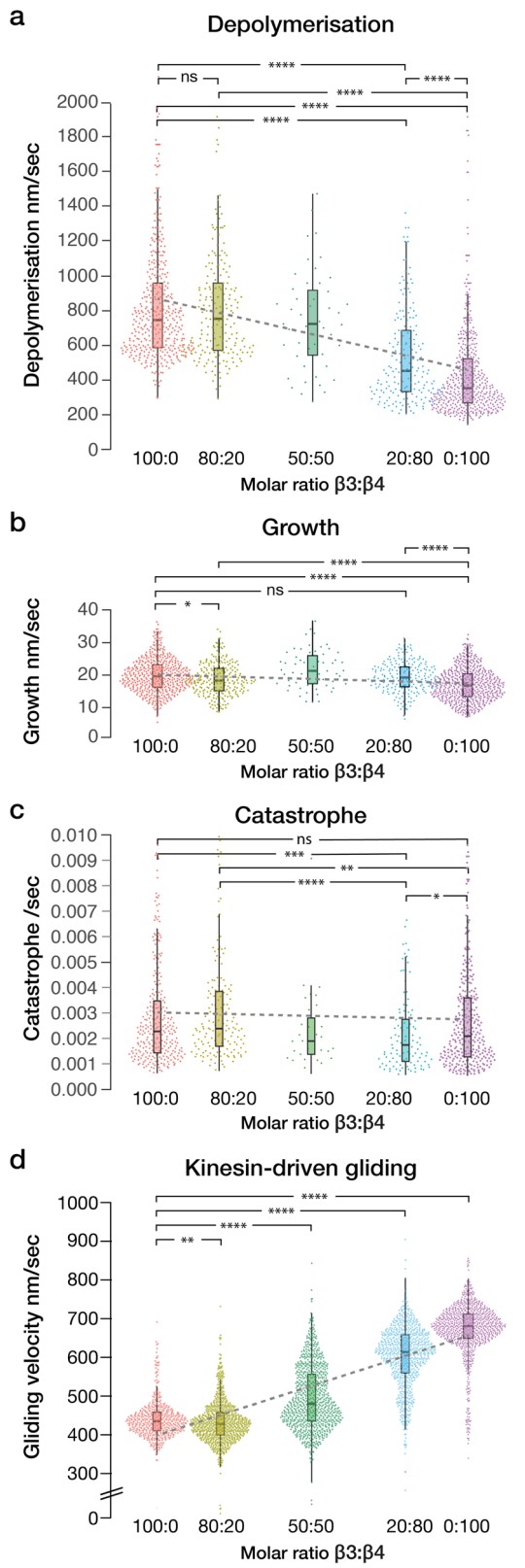

**Fig. 5 Lattice stability and gliding velocity of mosaic-isotype microtubules depend nonlinearly on tubulin isotype ratio. a** Plus end depolymerisation rate versus tubulin isotype ratio. Depolymerisation rates of microtubules with 100:0 Hs α1β3:Dr α1β4, 746.7 ± 324.6 9 (n = 404); 80:20, 756.4 ± 357.8 (n = 193); 50:50, 724.7 ± 287.2 (n = 47); 20:80, 453.3 ± 322.0 (n = 186); 0:100, 353.7 ± 277.5 (n = 467) nm/s (median ± SD). A straight line fitted by least squares to the raw data (broken line) is a poor fit. **b** Plus end growth rate versus tubulin isotype ratios, at 12 μM tubulin, except for the 50:50 ratio, which was at 15 μM tubulin. Growth rate of mosaic microtubules with 100:0 β3:β4, 19.4 ± 5.3 (n = 493); 80:20, 18.3 ± 5.1 (n = 234); 20:80, 19.3 ± 4.7 (n = 210); 0:100, 16.8 ± 5.0 (n = 551) nm/s. In this case, there is no evidence of nonlinearity, albeit the two growth rates are in any case similar. **c** Catastrophe rates. Rates for Hs α1β3 and Dr α1β4 are not significantly different. **d** Kinesin-driven gliding velocity in 10 μM taxol versus β3:β4 isotype ratio. Gliding velocity of mosaic microtubules with 100:0 β3:β4, 431 ± 49 (n = 505); 80:20, 424 ± 55 (n = 854); 50:50, 476 ± 82 (n = 845); 20:80, 609 ± 81 (n = 796) and 0:100, 672 ± 69 (n = 950) nm/s. A straight line fitted to the raw data by least squares is a poor fit, pointing to a nonlinear relationship. Single isoform data in (**a**–**c**) are from 3 experiments, 20:80 and 80:20 data are from 2 experiments, 50:50 data are from 1 experiment. Single isoform data in (**a**, **b**) is replotted from Fig. 1. In (**d**), β3 data are replotted from Fig. 4, which are from 2 experiments, others from 3 experiments. ns indicates p ≥ 0.05 and significance is *p < 0.05, **p < 0.01, ***p < 0.001 and ****p < 0.0001. For the box plots, boxes indicate interquartile range, whiskers mark SD, horizontal bar marks population median. All the data are plotted.

robust evidence that β3 tubulins within mixed isotype microtubules bind less taxol than other isotypes in the lattice[36]. Nonetheless it is clear that taxol in sufficient amounts can expand mixed isotype lattices that include β3 tubulins18, suggesting that β3 tubulins within a mosaic isotype lattice can be driven to expand via taxol binding to their neighbours. In this way, the affinity of β3 tubulin for taxol might be context-dependent, potentially leading to positive feedback effects in which local lattice expansion, induced by taxol, promotes further taxol binding and further expansion. By contrast in single isotype lattices, we hypothesise that α1β3 tubulins have a much lower taxol affinity than α1β4 tubulins, and so cannot achieve the higher taxol occupancy required for expansion.

In mixed-isotype brain microtubules at submicromolar taxol concentrations, <10% of taxol sites in the lattice are occupied[45]. It is unclear how these occupied sites are distributed. Taxol might bind unevenly between microtubules in the population. Along these lines, we find that with mixed isotype brain microtubules, while 10 μM taxol stabilises and accelerates the entire microtubule population, 200–400 nM taxol produces coexisting subpopulations of fast-gliding and slow-gliding stable microtubules (Supplementary Fig. 3). Further, taxol might bind unevenly at the level of individual microtubules. Taxol binding at microtubule tips might be favoured because the GTP-tubulin tip lattice is expanded relative to the core lattice (Fig. 6), and/or because the lumen of the microtubule is more accessible at microtubule tips. Preferential binding of certain fluorescent taxol analogues to tips and to lattice defects has been seen[39], but in our hands fluorescent taxols stabilise the GDP-lattice only marginally, which limits their value as reporters. Testing whether authentic, non-fluorescent taxol can bind the core GDP-tubulin lattice of α1β3 microtubules will require cryoEM imaging of α1β3-containing lattices at high resolution and high taxol concentrations. To the extent that taxol binds the α1β3 lattice, our data indicate that it does so without causing the go-faster conformational change.

Recent work indicates that taxol binds the GDP-tubulin lattice much more tightly than it binds to free tubulin because the

coupled. Prota et al.[22] recently showed that baccatin III, a taxol precursor, expands but only very marginally stabilises brain GDP-microtubules.

How does taxol stabilise and expand single isotype α1β4 lattices, yet stabilise single isotype α1β3 lattices without expanding them? The simplest hypothesis is that lattice expansion requires a higher taxol occupancy than lattice stabilisation (Fig. 6d). There is

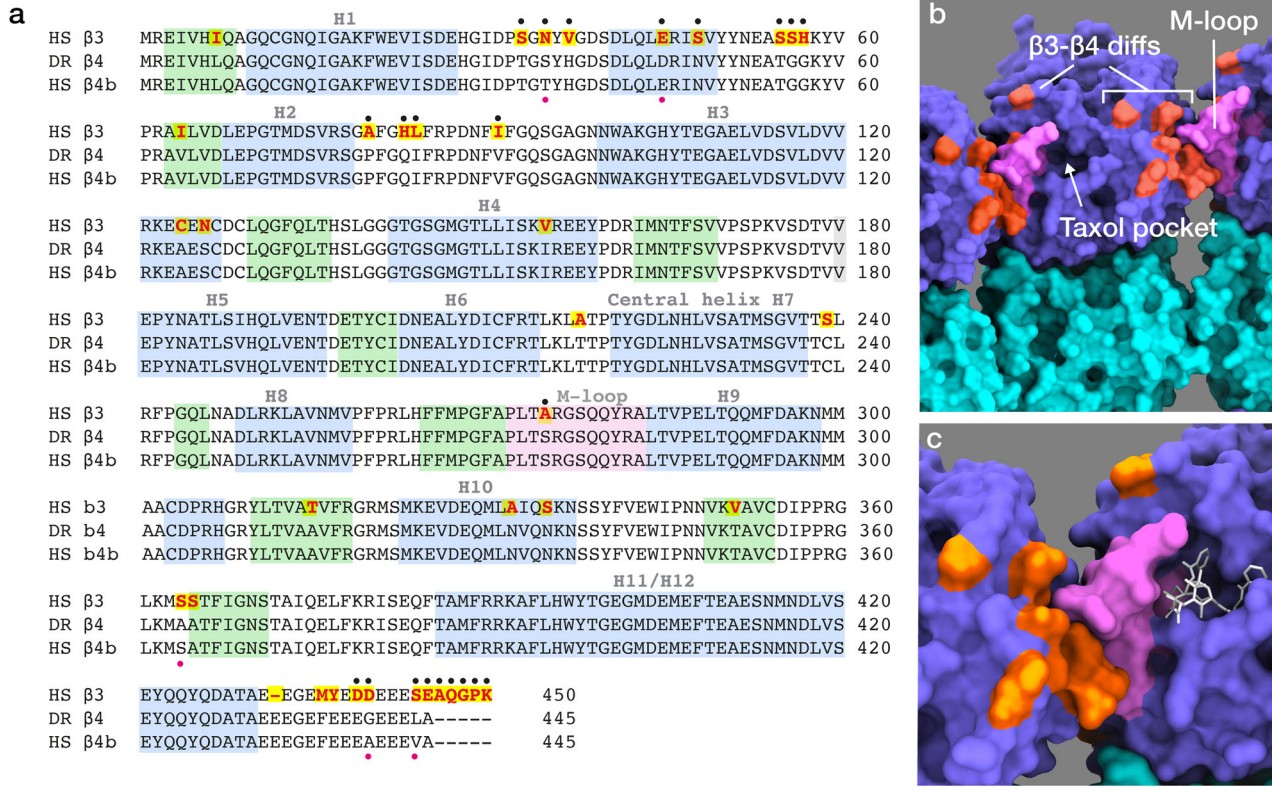

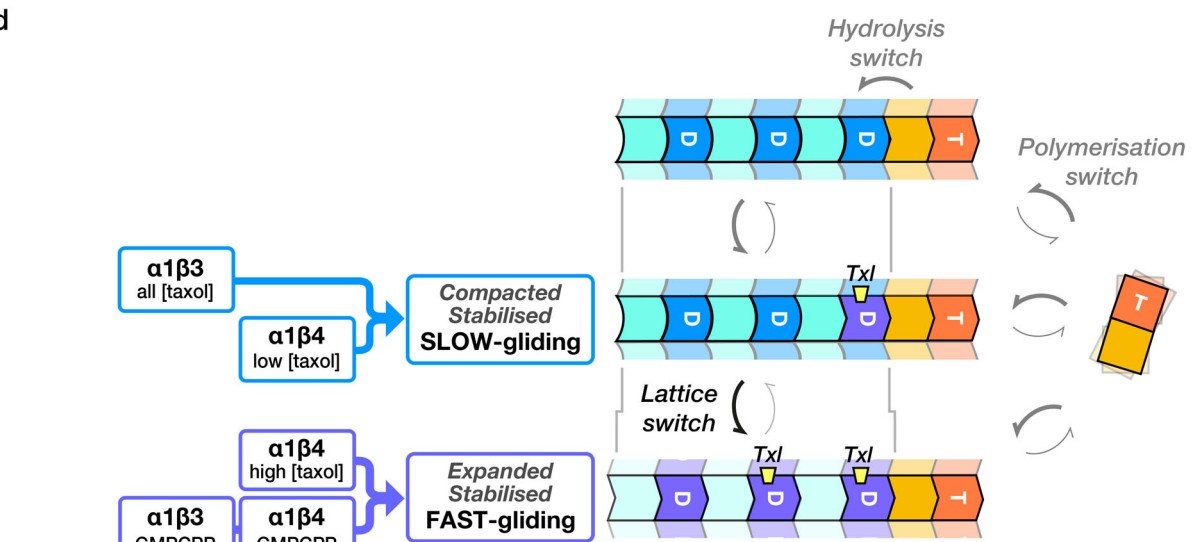

**Fig. 6 Structural-mechanical differences between β3 and β4 lattices. a** Sequences of human β3, human β4b and zebrafish β4 tubulins. Helices are shaded blue, sheets are green. Yellow highlighted red coloured residues differ between β3 and β4 isotypes. Black dots indicate the subset of these that is surface-exposed. The M-loop is shaded pink. Red dots highlight the 5 residues that differ between human and zebrafish β4 tubulins. Alignment done using Clustal Omega. **b** clustering of surface-exposed residue changes between β3 and β4 in the M-loop (pink) and in and around the M-loop interacting regions (orange, Supplementary Movie 7). **c** Zoom view of the M-loop region of a GDP-taxol subunit, same colours. Graphics made in Chimera X[57], using 6EVW.pdb. **d** Three-switch model. Incorporation of GTP-tubulin into the growing lattice shifts it into a lattice-compatible conformation (the polymerisation switch[46]). Hydrolysis and Pi release then convert lattice-resident GTP-tubulin to GDP-tubulin and introduce strain (the hydrolysis switch[6]). Strain in the GDP-tubulin lattice is relieved by taxol binding. Depending on the tubulin isotype, taxol binding (yellow wedges) stabilises the GDP-lattice in either a slow gliding/compacted (blue) conformation or a fast-gliding/expanded (purple) conformation. GMPCPP stabilises both β3 and β4 single isotype lattices in the fast-gliding/expanded (purple) conformation.

cytomotive polymerisation switch[46] (the tubulin conformational change induced by incorporation into the lattice) reconfigures the M-loop, which otherwise occludes the taxol site[22]. The lateral contacts made by the M-loops may be the dominant determinant of lattice stability[47]. The angles between neighbouring protofilaments fall as the protofilament number rises, and the resulting strain in the M-loops can be expected to affect their stability[48]. There is evidence that variations in protofilament number can influence the stability of mixed-isotype lattices[39]. We see multiphasic, often biphasic, depolymerisation of some α1β4

single-isotype microtubules (Fig. 1). This must reflect a structural difference between different regions of the depolymerising lattice, possibly differing numbers of protofilaments, flanking a defect. The M-loop sequence is conserved across all human β tubulin isoforms, except for the β3 and β1 (aka βVI) isoforms, both of which have an S275A substitution. β3 tubulin has an additional T218A substitution abutting the M-loop and the central helix H7 (Fig. 6a). Computational simulation suggested that T218A restricts the accessibility of the taxol pocket[36]. β3 tubulin also has non-consensus residues exposed in a cluster in those regions of its surface that engage the M-loop of the lateral neighbour in the lattice (Fig. 6b, c). Aside from changes in and around the M-loop itself, these are our mechanistic prime suspects, because restructuring and repositioning the M-loop can change the accessibility of the taxol site, the ability of the taxol site to signal its occupancy, and the basal stability of the lattice, seen as an ~2.2-fold difference in the depolymerisation rates of α1β3 and α1β4 GDP-tubulin lattices in the absence of taxol.

We worked predominantly with Dr β4 tubulin, but we included Hs β4 in some experiments. Dr and Hs β4 differ at only 5 residue positions, all surface exposed. 3 are in the regions that engage the M-loop of the lateral nearest neighbour and the other 2 are at the C-terminus (Fig. 6a). We expected Hs β4 and Dr β4 tubulins to behave alike, and they do have similar properties (Fig.1h–l). Remarkably however, we found that at 400 nM taxol, Dr β4 lattices appear predominantly expanded, yet Hs β4 appear predominantly compacted, as judged by their gliding velocities (Fig. 4a). This suggests that a very few apparently conservative residue substitutions may detectably shift the taxol-dependence of β4 lattice switching.

Between α1β3 and α1β4 tubulins, aside from differences in surface-exposed residues (Fig. 6a), there are further sequence differences in buried positions. We do not discount their possible influence. Substitution of certain buried residues can profoundly affect allosteric communication in β tubulin[49].

In segmented-isotype microtubules, taxol increases the difference in lattice stability between α1β3 and α1β4 segments. Neighbouring segments of different isotypes do not detectably affect each other's stability. Similarly, in gliding segmented-isotype microtubules, stabilised by 10 μM taxol, segments retain their characteristically different gliding speeds. Both sets of observations argue that the conformational properties of single-isotype lattice segments do not propagate axially. By contrast, local conformational signalling between neighbouring tubulin molecules within mixed-isotype lattices is clearly important. A 20:80 α1β3:α1β4 mosaic has significantly faster depolymerisation than α1β4 alone, whilst depolymerisation of an 80:20 α1β3:α1β4 mosaic is not detectably different to depolymerisation of a pure α1β3 lattice (Fig. 5a). The apparent cooperativity is striking and indicates that tubulin molecules in the lattice influence the conformations of their near neighbours. For the moment we are uncertain of the range of this allosteric signalling, because we cannot read the local isotype distribution in the lattice—we only know the isotype ratio in the assembly mix and biased recruitment of one isotype over the other is a possibility. In mosaic-isotype microtubules at different taxol concentrations, kinesin-driven gliding tends strongly to occur at either a fast or a slow rate (Fig. 5d), suggesting the lattice conformation of individual microtubules may shift in a concerted way.

We have used kinesin-driven microtubule gliding velocity as a sensor for lattice expansion, but how kinesin senses the conformational difference between α1β3 and α1β4 GDP-taxol lattices is unclear. The kinesin contact interfaces of α1β3 and α1β4 tubulins appear sequence-identical. The C-terminal tails (CTTs) are different, but experiments in which human β3 CTTs were substituted for the CTTs of Saccharomyces cerevisae tubulin

showed no effect of the human β3 CTTs on kinesin-1 velocity[50], consistent with the kinesin-1 stepping rate instead depending on the structured core[51] of tubulin (one CTT substitution, from β7 tubulin, did reduce kinesin velocity[50]). A tendency for kinesin-1 in vitro to enrich to brain microtubules with an expanded lattice has been reported[12]. Our working model (Fig. 6) posits that expansion couples to faster kinesin stepping and that taxol shifts the GDP-tubulin lattice between FAST/Expanded and SLOW/Compacted states dependent on the isotype and the taxol concentration. Faster binding of kinesin to an expanded lattice might contribute to faster processive stepping on taxol-stabilised α1β4 versus taxol-stabilised α1β3 microtubules. Does something like this happen in vivo? In neurons, kinesin-1 is a processive, axon-specific transporter of neurotransmitter vesicles. β3 tubulin is largely neuron specific. Recent work indicates that down-regulating human β3 tubulin boosts kinesin motility in neurons (without taxol)[52]. Perhaps relatedly, we see in vitro that a small decrease in the proportion of α1β3 tubulin in α1β3:α1β4 mixed-isotype microtubules substantially accelerates kinesin stepping in 10 μM taxol (Fig. 5d).

What might be the biological significance of isotype-specific conformational switching of the microtubule GDP-lattice? We think that, since tubulin isotypes freely copolymerise, the interesting problem is the extent and mechanism by which each isotype influences the properties of the composite lattice, both locally and globally. We speculate that a tipping point might occur, whereby the conformation and properties of the lattice switch over in a concerted (avalanche) manner when a particular proportion of one or another isotype is reached. If this happens, then switchability might be favoured by a mix of isotypes that accesses the tipping point. Particular MAPS might bind preferentially to a particular isotype within the lattice, so that certain MAPS might load preferentially at particular isotype ratios. Conversely, particular MAPS or sets of MAPS might bias the recruitment of particular tubulin isotypes into the lattice. The biological value of this might lie in the potential for feedback regulation of microtubule-based processes created by the co-dependence of isotype ratio, the recruitment of particular MAPS, and the tendency of the lattice to converge to a particular conformation/expansion state.

Our data provide insight into the lattice-mechanical action of taxol on different tubulin isotypes, and direct experimental support for earlier suggestions that β3 tubulin is poorly responsive to taxol, confirming that isotype switching in favour of β3 tubulin should help human tumour cells resist taxol[34]. The isotype composition of individual microtubules in vivo remains largely unknown, but the ability of isotypes to copolymerise is clearly established. Here, we have shown in vitro that taxol can change the conformation of one single-isotype lattice and not another, and that in mixed-isotype lattices, the GDP-lattice stability and kinesin-driven gliding velocity are both non-linearly related to the isotype ratio, suggesting taxol binding to a particular set of lattice-resident tubulin molecules can shift the conformation of their neighbours within the composite lattice. It will be important to explore the response of different tubulin isotypes to a wider range of microtubule-targeting drugs, including combinations of taxol and other microtubule-directed drugs[53]. Since taxol acts differently on different human tubulin isotypes, it may be that other tubulin drugs also have preferred isotypes and that this might be exploited therapeutically.

## Methods

**Constructs**. Individual sequence-optimised α and β tubulin genes (GeneArt), carrying an L21 leader sequence AACTCCTAAAAA ACCGCCACC at the 5′ to enhance protein expression, were

C-terminally joined to 8× his-(HHHHHHHH) and FLAG-(DYKDDDDK) affinity tags respectively with GGSGG inserted as a linker. The α tubulin genes are *Homo sapiens TUBA1B* (encodes NP_006073) and *Danio rerio tuba1c* (encodes NP_001098596). The β tubulin genes are *Homo sapiens TUBB3* (encodes NP_006077); *Homo sapiens TUBB4B* (encodes NP_006079) and *Danio rerio tubb4b* (encodes NP_942104). Supplementary Fig. 1 shows the sequences we used. Individual α and β tubulin genes were cloned into separate pLIB plasmids, both of which were subsequently integrated into one pBIG1 plasmid backbone (Supplementary Fig. 1) via Gibson assembly[54]. Briefly, forward and reverse primers flanked the gene of interest with respective extension sequences 5′-CCACCATCGGGCGCGGATCCA and 5′-TCCTCTAGTACTTCTCGACAAGCTT were used to amplify linear DNA from GeneArt templates. These linear α or β tubulin sequences were subsequently cloned into a pLIB backbone using Gibson assembly. Following successful transformation of Gibson assembly products, individual sets of primers, Cas I forward and Cas I reverse, Cas II forward and Cas ω reverse, were used to amplify the α and β tubulin sequences together with the polyhedrin promoter and transcriptional terminator sequences of the pLIB plasmids.

Primer sequences were as follows[54]:

Cas I forward AACGCTCTATGGTCTAAAGATTTAAATCG ACCTACTCCGGAATATTAATAGATC

Cas I reverse AAACGTGCAATAGTATCCAGTTTATTTAA ATGGTTATGATAGTTATTGCTCAGCG

Cas II forward AAACTGGATACTATTGCACGTTTAAATCG ACCTACTCCGGAATATTAATAGATC

Cas ω reverse AACCCCGATTGAGATATAGATTTATTTAA ATGGTTATGATAGTTATTGCTCAGCG

The resulting α and β tubulin constructs were then ligated into a pBIG1 backbone using Gibson assembly. The pBIG-α-β plasmid (Supplementary Fig. 1) was then amplified in *Escherichia coli* DH5α cells, miniprepped and integrated into the baculovirus genome via Tn7 transposition sites in *E. coli* DH10EmBacY to allow subsequent generation of recombinant baculovirus. Clones with successfully integrated baculovirus were selected using appropriate antibiotics and blue-white screening. The recombinant virus DNA bacmid was isolated from overnight culture using MidiPrep (Qiagen) reagents (without the use of a column). The DNA was precipitated overnight in absolute isopropanol at −20 °C, pelleted the next day at $14,000 \times g$ for 15 min, washed in 70% ethanol and pelleted again. The pellet was air-dried and dissolved in water. The isolated bacmid DNA was stored at 4 °C. Successful insertion of tubulin genes was verified by PCR using M13 forward and reverse primers.

**Baculovirus generation**. Sf9 cells were used for recombinant baculovirus generation and protein expression. Cells were maintained at 28 °C in Ex-Cell 420 media (Merck), with shaking at 120 rpm for suspension culture and/or grown as static adherent cells. Generally, cultures were split when they reached a density of approximately $2.0 \times 10^6$ cells/mL to achieve $0.5 \times 10^6$ cells/mL.

For P1 virus production, 2 mL of $0.5 \times 10^6$ cells/mL were seeded onto a 35 mm petri dish and allowed to stand for 15 min for cell adherence. Transfection mix was prepared by adding 2 µg of bacmid DNA into 200 µL of fresh media, followed by addition of 6 µL of FuGene HD (Promega). This entire mix was dripped onto the culture and incubated without shaking. P1 virus was harvested, typically after 3–5 days of incubation, after clarification by centrifugation at $750 \times g$ for 5 min at room temperature. P2 virus was generated using one volume of P1 virus added into 100 volumes of fresh culture at a density of $1.0 \times 10^6$ cells/mL. P2 virus was harvested 3 days afterwards, as for P1 virus, except the supernatant was filtered through a 0.45 µm PVDF syringe filter.

P3 virus, which was used for tubulin expression, was generated similarly to P2 virus but 1 volume of P2 virus was used to infect 50 volumes of culture instead. All passages of virus were kept at 4 °C for storage, away from light.

**Buffers**. Recombinant tubulin purification, microtubule assembly and microtubule dynamics assays were based on KPEM buffer (100 mM PIPES, 2 mM EGTA, 1 mM MgSO$_4$, pH adjusted to 6.9 with KOH pellets), supplemented with other components as detailed in other sections. BRB 80 buffer (80 mM PIPES, 1 mM EGTA and 1 mM MgCl$_2$, pH adjusted to 6.8 with KOH pellets) was used in microtubule gliding assays, with other components added at different stages as shown in the gliding assay section.

Porcine brain tubulin purification required depolymerisation buffer (DB) and high-molarity PIPES buffer (HMPB), as well as BRB 80. DB was 50 mM MES, 1 mM CaCl$_2$, with pH adjusted to 6.6 with HCl. HMPB was made up of 1 M PIPES, 10 mM MgCl$_2$, and 20 mM EGTA, adjusted to pH 6.9 with KOH pellets. All buffers were filtered through a 0.2 µm filter.

**Tubulin expression and purification**. Tubulin expression and purification is based on ref. [38]. In detail, we did as follows. 30 mL of P3 virus was added into 1 L of Sf9 culture at a cell density between 1.5 and $2.0 \times 10^6$ cell/mL. Cells were harvested after about 56 h of incubation at 28 °C with shaking at 120 rpm. Cell pellets were collected by centrifugation at $750 \times g$ at 4 °C for 20 min. The pellets were gently resuspended in ice-cold PBS and the cells were collected again by centrifugation. These washed pellets were then flash-frozen in liquid nitrogen and stored at −80 °C. On the day of purification, all buffers used were supplemented with a final concentration of 1 mM ATP and 1 mM GTP, unless otherwise stated. All purification steps were performed on ice or at 4 °C. Cells were thawed then lysed in an equal volume of KPEM buffer supplemented with 0.5 M 3-(1-pyridinio)-1-propane sulfonate, ~25 unit/mL Benzonase® nuclease, 1 mM DTT, 1 mM PMSF, 0.05% CHAPS, 25 mM imidazole and 1% glycerol. The mixture was homogenised with a douncer using about 60 strokes, then clarified by centrifugation at $300,000 \times g$ (T865 rotor) for 45 min. The supernatant was loaded on to a 5 mL HisTrap HP (Cytiva) at 2 mL/min. This column was washed with 20 column volumes of KPEM supplemented with 25 mM imidazole, 250 mM KCl, and an additional 2 mM MgSO$_4$. The protein was then step-eluted with similar buffer but made up to 400 mM imidazole. The pooled fractions were diluted with an equal volume of KPEM buffer to achieve a final concentration of 125 mM KCl and then incubated for 1 h with 4 mL anti-FLAG monoclonal antibody-conjugated resin (M2 agarose, Merck) using rolling. The resulting resin was packed into a 10 mL column, washed with 5 column volumes of KPEM supplemented with 125 mM KCl, and finally eluted with 5 column volumes of 250 µg/mL FLAG peptide in the wash buffer. The pooled eluate was diluted 4 times with KPEM buffer to reduce the KCl concentration to about 30 mM and applied to a 1 mL Capto HiRes Q 5/50 GL column (Cytiva). This column was then washed with KPEM and eluted with KPEM supplemented with 0.5 mM GTP and 0.5 mM ATP, 300-400 mM KCl (isotype dependent). The pooled eluate was exchanged into buffer KPEM without nucleotide using a HiPrep 26/10 desalting column (Cytiva) and finally concentrated with an Amicon 30 kD regenerated cellulose spin concentrator to at least 40 µM. Tubulin was snap-frozen in liquid nitrogen in 20 µL aliquots and stored in liquid nitrogen. Aliquots were removed freshly before each experiment. Concentrations were determined using a spectrophotometer (Cary 50) with molar extinction coefficient 107,110 M$^{-1}$ cm$^{-1}$ for both *Danio rerio* α1cβ4b and *Homo sapiens* α1bβ4b tubulins and 108,390 M$^{-1}$ cm$^{-1}$ for *Homo sapiens* α1bβ3 tubulin at wavelength 280 nm. Supplementary Fig. 2a shows the purity of the tubulins.

Supplementary Fig. 2b shows representative steps in the purification of Hs α1β3 single-isotype tubulin. The yield of recombinant tubulin is about 0.75 mg/L of Sf9 culture.

Porcine brain tubulin was purified broadly according to ref. [55], using cycles of polymerisation, centrifugation and depolymerisation. Polymerisation was performed at 37 °C in 1:1:1 volume ratio of tubulin:HMPB:glycerol whereas depolymerisation was carried out at 4 °C in DB, unless otherwise stated. In detail, we did the following. Porcine brains were freshly harvested from the slaughterhouse on the day of purification, transported back to the lab on PBS ice cubes, taking 1–2 h. Blood clots were then removed from the brains, which weighed about 300 g. The brains were blended together with cold DB at a ratio of 1 litre of buffer for every 1 kg of brains. The mixture was spun down at $29,000 \times g$ for 60 min in a SLA 1500 Sorvall rotor at 4 °C. The supernatant was then transferred into a vessel together with one volume of glycerol and one volume of HMPB, both of which were pre-warmed to 37 °C. This mix was supplemented to a final concentration of 1.5 mM ATP and 0.5 mM ATP and incubated for 80 min at 37 °C. The polymerised tubulin was then spun down at $151,000 \times g$ in a Ti 45 Beckman rotor for 30 min at 37 °C. The microtubule pellets were gently homogenised with a douncer in 100 mL of DB and left on ice for 45 min to depolymerise. Following this depolymerisation step, the tubulin was clarified at $70,000 \times g$ for 30 min at 4 °C in a Ti 45 rotor. The tubulin was then again polymerised in 1:1:1 tubulin:HMPB:glycerol with ATP and GTP at 37 °C as above for 30 min. The resulting microtubules were again pelleted down. The microtubule pellets were resuspended in 4 mL of ice-cold BRB 80, incubated on ice for 15 min. After clarification at $100,000 \times g$ at 4 °C for 10 min, the supernatant was loaded on to a HiLoad 26/60 Superdex 200 prep grade column (Cytiva) for overnight gel filtration chromatography at 4 °C at a flow rate of 0.5 mL/min. Peak fractions from this gel filtration step were pooled, resuspended in BRB 80 buffer, supplemented with 10 nM GTP, flash-frozen in liquid nitrogen in 15–30 μL aliquots, and stored in the vapour phase of a liquid nitrogen Dewar. Tubulin concentration was determined by absorbance at 280 nm wavelength with molar extinction coefficient of $105,838\ \mathrm{M}^{-1}\,\mathrm{cm}^{-1}$. Before use, tubulin aliquots were quickly thawed in between hands, and kept on ice until right before use. Unpolymerised tubulin, stored on ice, was used within 1 h.

**Kinesin expression and purification.** *Drosophila* full-length kinesin-1 (gene *Khc*; NP_476590) was C-terminally tagged with 6x his sequence with a thrombin cleavage site in between[56]. Successfully transformed *E. coli* BL21 pLysS cells were maintained at 24 °C with shaking at 180 rpm. Due to leaky protein expression, addition of IPTG was omitted. This led to a better yield of soluble protein in our hands. The culture was harvested after reaching an optical density of 0.9 at 600 nm. Kinesin purification was performed at 4 °C or on ice unless otherwise stated. The culture was centrifuged at $15,000 \times g$ for 6 min. The pellet was gently resuspended in ice-cold PBS and again pelleted. This second pellet was suspended in 3× its mass of a buffer containing 10 mM Tris-base, 4 mM Mg-acetate, 250 mM potassium acetate, 1 mM TCEP, supplemented with 100 μM ATP, 1× complete protease inhibitor (Roche), 0.5% vol/vol Triton X-100, ~125 unit/mL Benzonase® Nuclease, 0.1 mg/mL lysozyme, with pH adjusted to 8.0 with acetic acid. The bacteria were lysed by sonication in this buffer (Misonix, S-4000 Ultra-sonicator) with 6 cycles of 10-s pulses at 35 Amps, followed by 30-s cooling after every round. This whole cell lysate was clarified at $20,000 \times g$ for 20 min and the supernatant was passed through glass wool. The clarified supernatant was then applied to a 1 mL HiTrap Talon crude column (Cytiva) at 1 mL/min and eluted with 100–150 mM imidazole in 10 mM Tris-base, 4 mM magnesium acetate, 250 mM potassium acetate, supplemented with 1 mM TCEP and 100 μM ATP. The pooled affinity-purified eluates were then further purified by application to a 1 mL HiTrap Q HP column (Cytiva) followed by elution with 200 mM NaCl in 10 mM Tris-base, 4 mM magnesium acetate, 250 mM potassium acetate, supplemented with 1 mM TCEP and 100 μM ATP. The pooled eluates were mixed with glycerol to 20% glycerol and snap-frozen in liquid nitrogen for storage.

**Glass slide treatment.** To produce a hydrophilic surface, microscopic slides (75 mm × 26 mm × 1.0 mm, Thermo Scientific) and coverslips (no. 1.5, 22 mm × 22 mm or 22 mm × 50 mm, Thermo Scientific) were sonicated in 3% Neutracon in ultrapure water (18 MOhms) for 30 min, followed by 11 rounds of sonication and rinsing. The cleaned coverslips were stacked together and stored in a zip-lock bag to minimise contact with air.

**Flow cell assembly.** For dynamics assays, flow cell channels were assembled using two thin strips of double-sided tape (3 M Scotch tape), about 3 mm apart, sandwiched between a glass slide (75 mm × 26 mm × 1.0 mm) and a coverslip (no. 1.5, 22 mm × 22 mm). The channel volume was about 10 μL. For gliding assays, flow cell channels were assembled from two coverslips (no. 1.5, 50 mm × 22 mm and 22 mm × 22 mm) with double-sided tape in between as a spacer. Each channel was about 5 μL. The channels/surface could be subsequently treated as desired by flowing in different solutions. Solution exchange was achieved by introducing new solution with a micropipette from one end of the channel and drawing out the existing solution from the other end using grade 1 Whatman filter paper. To ensure maximal solution exchange, new solutions were introduced by flowing in 5 flow channel volumes.

**Glass surface passivation for dark-field imaging of dynamics assays.** Pre-cleaned glass slides/coverslips (above) were plasma-cleaned (Air plasma, Henniker plasma HPT-200) for 5 min immediately before assembly into a flow cell. Each channel was filled with 0.2 mg/mL of PLL-PEG-biotin (SuSoS) in PBS for 30 min, washed out with KPEM, then incubated with 1 mg/mL NeutrAvidin (Thermofisher Scientific) in KPEM buffer, and washed again with KPEM. Biotinylated microtubule seeds were then introduced.

**Microtubule GMPCPP seed preparation.** To allow one end of each microtubule to be anchored to the glass surface, short segments of microtubules called seeds were used. Tubulin for seeds was prepared using 26 μM of porcine brain tubulin containing 10% biotinylated tubulin and 10% HiLyte 488 tubulin (both from Cytoskeleton) in 1 mM GMPCPP (Jena Bioscience) in KPEM. This mixture was incubated on ice for 15 min, clarified using an AirFuge (A-110 fixed angle rotor, Beckman Coulter) to remove aggregates, flash-frozen in liquid nitrogen and stored over liquid nitrogen. To polymerise tubulin into seeds, an aliquot of seed assembly mixture was thawed and incubated at 37 °C for 30 min. Residual free tubulin was removed by centrifugation at 20 psi in an AirFuge for 5 min at room temperature. Pelleted seeds were gently resuspended by aspiration and diluted in KPEM to obtain optimal decoration of the flow cell channel. Flow cells were passivated with PLL-PEG-biotin as above. Unbound seeds were washed out from flow chambers using 1% tween 20 in KPEM. Tubulin mix for dynamics assays was then introduced.

**Microtubule dynamics assay.** Tubulin at different concentrations (with or without taxol) was prepared in KPEM imaging buffer with an oxygen scavenger system (4.5 mg/mL glucose, 0.2 mg/mL

glucose oxidase, 35 µg/mL catalase and 0.5% (v/v) β-mercaptoethanol (GOC)), 1 mM GTP, 1 mg/mL BSA and 0.2% tween 20. All solutions were freshly clarified by AirFuge at 4 °C for 5 min to remove large particles which might interfere with darkfield imaging. Tubulin mix was introduced into flow channels pre-decorated with GMPCPP seeds and dynamics were observed at 30 °C. To prevent evaporation, flow cell ends were sealed with vacuum grease after tubulin mix was introduced into the seed-decorated flow cell channels.

**Preparation of segmented-isotype microtubules for dynamics assays.** Segmented microtubules for dynamic instability assays were assembled in the presence of 100 nM taxol. Porcine brain GMPCPP-microtubule seeds were immobilised in flow cells using neutravidin- binding of the biotinylated tubulin included in the GMPCPP seeds. An initial segment of α1β3 tubulin was grown from these immobilised seeds by flowing in 12 µM single-isotype α1β3 tubulin. Following incubation at 30 °C for ~15 min, α1β3 tubulin was flushed out with α1β4 tubulin and a second segment of α1β4 tubulin grown. Microtubules were imaged both during the growth of each segment and following taxol washout. To observe microtubule depolymerisation at different taxol concentrations, free tubulin was washed out with taxol solution at a defined taxol concentration. Depolymerisation rates were obtained upon restraightening of microtubules after flow had stopped.

**Darkfield/epifluorescence microscopy.** Darkfield Images were acquired with an electron-multiplying charge-coupled device (EMCCD) camera (Andor, iXon DU 897) fitted to a Nikon E800 microscope with a ×100 objective (Plan Fluor NA 0.5–1.3 variable iris, Nikon). Samples were illuminated with a 100 W mercury short-arc lamp (102D, Ushio) connected to the microscope with a fibre optic light scrambler (Technical video) and the unwanted light was filtered out with a cold mirror and green interference filter with 500–568 nm band pass (Nikon). Epifluorescence was achieved using a stabilised mercury lamp (X-cite exacte, Lumen Dynamics) with light pipe connection to the microscope. Microscope temperature was maintained at 30 °C using a custom-made chamber and a heater (Air-Therm ATX, World Precision Instruments). An electronic shutter was used to switch between fluorescence and darkfield illumination. Microscope and camera were controlled by Metamorph software (Molecular Devices). For dynamics experiments, 300 ms exposure and 200 ms exposure time were used for darkfield and epifluorescence respectively, taking 1 frame of epifluorescence for every 300 frames of dark-field illumination, with 1 s frame intervals and 160 nm per pixel.

**Microtubule preparation for gliding assays.** Single isoform or porcine brain or mixed isoform tubulin was mixed with 5% (molar ratio) HiLyte 488 or HiLyte 647 conjugated porcine brain tubulin. HiLyte tubulin (Cytoskeleton) was reconstituted to 40 µM in buffer KPEM, clarified with the AirFuge at 4 °C for 5 min to remove aggregates and snap-frozen in small aliquots for storage.

Microtubules for gliding assays were assembled at 40 µM tubulin in 1 mM GTP in KPEM for 30 min at 37 °C. The reaction mix was then diluted with an equal volume of taxol solution supplemented with 1 mM GTP to achieve a final concentration of 20 µM tubulin, with taxol at 10 µM or 400 nM, and further incubated for 30 min. For microtubule gliding assays performed at 200 nM taxol, microtubules assembled at 400 nM taxol were used instead, as very few microtubules were formed at 200 nM taxol.

GMPCPP tubulin mix was prepared by incubating 10 µM single isotype tubulin (containing 5% fluorescently labelled porcine brain tubulin) with 1 mM GMPCPP on ice for 15 min, clarified by AirFuge centrifugation as above to remove aggregates and snap-frozen in small aliquots. For assembly, the tubulin mix was quickly thawed, incubated at 37 °C for half an hour, then supplemented with equal volume of 1 mM GMPCPP in buffer KPEM and incubated for an hour at 37 °C.

**Preparation of segmented isotype microtubules for gliding assays.** Each type of single isoform microtubules were assembled as previously described in 10 µM taxol, pelleted using brief Air-fuge centrifugation to remove free tubulin and resuspended by aspiration in KPEM supplemented with 10 µM taxol. The mixture of two types of single isoform microtubule was incubated at room temperature overnight to allow spontaneous end-to-end joining.

**Microtubule gliding assay.** Gliding assays were performed at room temperature (23–25 °C) on glass surfaces coated with either 1% nitrocellulose in amyl acetate, by dripping the solution onto a lens tissue and wiping across the surface, or coated with 0.2 mg/mL α-casein in BRB 80 buffer supplemented with 1 mM DTT, before introducing kinesin into the channel. Kinesin solution supplemented with 1 mM DTT and 0.1 mg/mL α-casein in BRB 80 buffer was introduced into the channel and incubated for 5 min to allow adsorption to the surface. Subsequently, unbound kinesin was washed out with 5 channel volumes of BRB 80 supplemented with 1 mM DTT and 2 mM ATP (together with or without 2 mM ADP for segmented microtubules). Suitably diluted microtubules, typically about 30–50× dilution in gliding buffer BRB 80, supplemented with 1 mM DTT and 2 mM ATP (with or without 2 mM ADP for segmented microtubules) and 1× GOC with corresponding concentration of taxol, was introduced, briefly incubated to allow sufficient microtubule density. Unbound microtubules were washed out with gliding buffer with or without taxol according to the experiment. Fluorescent microtubules were visualised using an eduWOSM microscope. The eduWOSM uses a miniature 4-colour LED light engine and a 100x Nikon oil immersion objective. Please see https://wosmic.org/projects/eduwosm/Index.php for details. Time series were recorded typically at one frame every 5 s, taking 200 ms exposure per frame.

**Statistics and reproducibility.** Data described as being from several experiments are from repeats of the same experimental procedure using freshly thawed samples of the same proteins on different occasions. Kymographs were generated in FIJI using the "MultiKymograph" plug-in, by overlaying an 11-pixel straight-line ROI on microtubules. Growth and depolymerisation rates were determined by fitting a linear ROI over the tip of each microtubule in the kymograph by eye. For multisegmented depolymerisation events, each segment of the ROI was treated as one data point ($n = 1$), unless otherwise stated. Microtubules with length change <1 µm per fitted ROI segment were not considered for quantification for growth and depolymerisation rates. For microtubule gliding assays, the Fiji Plugin MTrackJ was used to track the tips of advancing microtubules, taking the angular change, "Δθ", due to the most recent displacement vector (the vector pointing from the previous point to the current point of the track). For measurements of gliding velocity, each instantaneous velocity (determined between two consecutive frames) was deemed as one data point ($n = 1$), unless otherwise stated. For general statistical analysis we used a Mann–Whitney U-test. For the cumulative density function analysis in Fig. 4j we used a two-sample Kolmogorov–Smirnov test. For both tests, ns indicates $p \geq 0.05$ and significance is *$p < 0.05$, **$p < 0.01$, ***$p < 0.001$ and ****$p < 0.0001$. Fitting of Gaussian distributions was done using

R with function non-linear least squares (nls). Where box plots are used, boxes indicate interquartile range, whiskers mark SD, horizontal bar marks population median. All the data are plotted.

**Reporting summary**. Further information on research design is available in the Nature Portfolio Reporting Summary linked to this article.

## Data availability

All data generated or analysed during this study are included in this published article [and its Supplementary Information files]. Source data for figures can be found in Supplementary Data 1 and any remaining information can be obtained from the corresponding author upon reasonable request.

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

## Acknowledgements

We thank Carolyn Moores for insightful comments, Thilani Babuji for technical support, Huong Vu for help with data analysis and fellow members of the Cross lab and the CMCB for improving this manuscript. R.A.C. dedicates this work to Linda Amos, who is much missed. Funded by a Wellcome Investigator Award to R.A.C. [220387/Z/20/Z]. Y.M.C. was funded by a PhD studentship from the MRC DTP in Interdisciplinary Biomedical Research awarded to University of Warwick (grant number MR/N014294/1) by the Medical Research Council.

## Author contributions

Y.M.C. carried out all experiments and data analysis. Y.M.C. and R.A.C. designed and interpreted experiments together and wrote the manuscript together.

## Competing interests

The authors declare no competing interests.
