## [Peer Review File · Communications Biology]

Reviewers' comments:

Reviewer #1 (Remarks to the Author):

In their paper, Chew and Cross use in vitro reconstitution with purified components to tackle multiple questions in the microtubule field about the functions of taxol, lattice expansion and compaction, and tubulin isotype differences. One of the most fundamental parts of this paper is that the authors are able to differentiate between the stabilizing and expanding functions of taxol using two tubulin isotypes: $\alpha\beta4$ and $\alpha\beta3$. They find that $\alpha\beta4$ lattices are both stabilized and expanded by taxol, while $\alpha\beta3$ lattices can only be stabilized by taxol (and only at much higher concentrations). They perform a functional assay with kinesin-1 and find that taxol accelerates kinesin gliding of $\alpha\beta4$ lattices, but not $\alpha\beta3$ lattices. This is consistent with this lab's prior work showing kinesin-1 can actively expand the GDP lattice, but further implies kinesin-1 will preferentially bind an already expanded lattice over a compacted lattice.

The study is well performed, the results are exciting, and the discussion is engrossing. My only suggestions are 1) to include a reference to our recent work showing that tau can compact the lattice (Siahaan et al., 2022) within the first paragraph of the introduction when the authors list EB1 and doublecortin, and 2) to expand a bit in the discussion about other implications of their work. For example, do the authors hypothesize that $\alpha\beta3$ lattices are resistant to expansion? I find this to be an intriguing possibility, but the authors have more knowledge on this topic than I. It could be that certain lattices are kept in either an expanded or compacted state due to MAPs that prefer one state over the other. OR, it could be that certain lattices are capable of being expanded and certain lattices are simply not. Also, what do the authors think about the physiological purpose of these different isotypes within different cell types. The authors hint about $\beta3$ tubulin being highly abundant in neurons and that downregulating this isoform boosts kinesin-1 motility. Would you speculate that $\beta3$ lattices are preferred by other motors and that non- $\beta3$ lattices exist specifically for kinesin-1? Or, since kinesin-1 can obviously still associate with $\alpha\beta3$ lattices, is this a mechanism to control the speed of kinesin-1-driven cargo in particular? If the authors don't feel comfortable speculating, I understand (though I would still like to read their thoughts in response to this review). It's a great study that should be published regardless.

Reviewed by Cassandra Ori-McKenney

Reviewer #2 (Remarks to the Author):

I read with interest the manuscript by Chew and Cross concerning the differential effect of Taxol on tubulin isotypes. I found the manuscript of interest, timely and I feel that it can be published in Communications Biology after revision.

My main concern is the fact that the interpretation of the observations relies on two hypothesis that does not seem to be supported by bibliography. First, individual tubulin subunits in the lattice can be either in the expanded or the compressed state, therefore lattices can be inhomogeneous. And second, $\beta3$ tubulin subunits do not expand upon Taxol binding. Authors should consider the evidences supported by others references to build up a more suitable hypothesis considering their results.

They mention Ref. 35 and 18 in their manuscript. Ref 35. (which is now eLife 12:e84791) studies the effect of Taxol, docetaxel and other taxanes on microtubule lattice using fiber diffraction technique and purified calf brain tubulin. Fiber diffraction measurements rely on diffraction selection rules for helical structures, which depend on helical samples with a homogenous pitch. When several layer lines are observed it is because the microtubules have a homogeneous lattice. In these experiments authors used calf-brain tubulin that roughly contains 33% of $\beta3$, 50% of $\beta2$ and likely about 10% of $\beta4$. Their images show a homogeneous expansion of the microtubule lattice upon occupancy of the taxane site indicating that $\beta3$ tubulin isotype is able to expand.

And this is not the only evidence because similar results are also shown in reference 18. Here, microtubules are assembled from porcine brain tubulin (similar isotype content) and, are also expanded when bound to Taxol. A final observation from reference 18, which is opposite to previous hypothesis, stated that microtubules require to be fully bound to Taxol to expand because in the presence of taxol:tubulin ratios lower than 1, as it is in this manuscript, microtubules keep a compressed lattice. Of course, this bulk solvent technique does not allow to identify local expansions.

Finally, data from Estevez Gallego et al (2020) 10.7554/eLife.50155 (not referenced in this manuscript and with experiments done in a similar way than ref. 18 and 35), shows that Taxol induced lattice expansion can be reverted by the occupancy of the β -phosphate site with phosphate analogs.

Taxol to tubulin ratios employed in this work are far below those employed in the references mentioned above and, therefore local expansion in certain parts of the microtubule (kinks) might be possible. Also, it is possible that neighbor subunits have a strong influence in the conformation of a certain tubulin dimer. Considering their results and literature evidences it seems more plausible that the differential effect of Taxol on $\beta 3$ and $\beta 4$ tubulin isoforms is related to differences on binding affinities of the drug to these. Of course, it might also be possible that $\beta 3$ subunits can only expand when in inhomogeneous lattice due to the effect of neighbor subunits. However, in the absence of other structural data I find the claim that $\beta 3$ tubulin subunits cannot expand is an overestimation of their results and, contradictory with previous structural studies.

In summary, I find that, in order to the manuscript be accepted, the authors should first, soften the specific claim that $\beta 3$ tubulin subunits cannot expand. Second, they should consider that Taxol can have different affinities for $\beta 3$ and $\beta 4$ subunits, which could in turn produce other lattice conformational changes different to expansion (e.g. lateral angle between protofilaments that could also affect kinesin motility). And third, they should discuss the apparent contradiction with previous structural results.

Minor concerns.

Why does the authors select $\beta 4$ tubulin and not the more common $\beta 1$ and $\beta 2$ isoforms?

I wonder why the authors co-assembly zebrafish $\beta 4$ tubulin with human $\beta 3$ tubulin instead of sticking to human proteins (especially considering the differences between both $\beta 4$ isoforms found, Figure 4a)

Note that the comparison between GMPCPP stabilized microtubules and Taxol stabilized microtubules is not rigorous because in the conditions of the study GMPCPP stabilized microtubules are homogeneous (they have 100% GMPCPP bound subunits) while taxol stabilized microtubules have only a percentage of tubulin molecules bound.

Note that Taxol bound and unbound microtubules are not only different in the axial direction but also in the lateral direction (different number of protofilaments) Ref 35 and 10.1016/j.str.2022.11.006 and that this change is extremely easy and quick 10.1074/jbc.273.50.33803 this should be considered as well when discussing the slow and fast gliding lattices.

Line 40. Reference 2 is not related to this sentence.

Line 41. Consider citing Estevez Gallego et al (2020) 10.7554/eLife.50155 for the effect of γ -phosphate analogs and GMPCPP vs GMPPCP.

Line 46. Colchicine was the first tubulin directed drug to be discovered, and Vinca alkaloids the first in clinical use.

Lines 48-51. Consider to reference these sentences.

Line 54. Ref 35 describe crystallographically precise molecular mechanisms of microtubule stabilization of taxanes.

Line 63. Ref 20 does not show that Taxol can assemble GDP-tubulin molecules, only that it does not require external GTP. Consider citing Díaz and Andreu *Biochemistry* 32, 2747-2755 (1993) Assembly of Purified GDP-Tubulin into Microtubules induced by Taxol and Taxotere: Reversibility, Ligand Stoichiometry, and Competition.

Line 128. From d. Should be from e

Line 131: μ : mean; I miss the units of μ

Line 230: There is no figure 4i!

Lines 385-387: References 39 and 40 are the same.

Line 593: Hope the channel volume is 10 μ l not 10 ml

Line 597: The same as 593

Line 664: 40 mM tubulin is 4.4 g/ml seems a bit too much!

Line 677: 10 mM tubulin is 1 g/ml in general PLEASE CHECK THE UNITS ALL ALONG THE MANUSCRIPT

Reviewer #3 (Remarks to the Author):

This study by Chew et al examines whether tubulin isotypes exhibit different rates of microtubule dynamics and sensitivity to the drug, taxol. These are important questions in the microtubule field. We have known about tubulin isotypes for several decades, but whether isotypes exhibit distinct activities and drug sensitivity has remained incompletely understood. This question is particularly important for the chemotherapeutic drug, taxol. Taxol resistance is associated with aberrant isotype expression, but we do not understand how isotypes determine taxol sensitivity.

To address these questions, the authors use recombinant approaches to generate isotypically pure samples of tubulin. They use these samples to show that heterodimers containing β 3 tubulin exhibit faster depolymerization than heterodimers containing β 4 tubulin. Furthermore, β 3 heterodimers maintain microtubule dynamics at concentrations of taxol that dampen the dynamics of β 4 tubulin. The authors provide evidence that taxol generates a lattice state in microtubules assembled from β 4 tubulin that supports faster rates of kinesin motility; however, taxol fails to generate a similar lattice state in microtubules assembled from β 3 tubulin. This is an important finding that establishes a new mechanistic explanation for how β 3 expression levels may determine cellular response to taxol treatment. The authors also show that this effect exhibits a non-linear relationship to the concentration of β 3 tubulin in the microtubule, suggesting that β 3 may influence the structure of neighboring tubulins through an allosteric mechanism.

The study addresses important questions using powerful and recently developed techniques. While some of the results confirm published findings by other labs, the results here go further toward a unified view of how isotype composition might propagate distinct states of the lattice that could influence microtubule dynamics, kinesin activity, and drug response. While we are enthusiastic about the study, there are weaknesses in the design, interpretation and writing of the current manuscript that must be amended prior to publication. Addressing these concerns would improve clarity and strengthen the conclusions of the study.

Major concerns:

1) A central weakness of the study is that the authors do not measure taxol binding to microtubules containing β 3 vs β 4 heterodimers. This is an important point because the apparent insensitivity of β 3 tubulin to levels of taxol that are sufficient to strongly affect β 4 could be due to isotype-specific differences in binding affinity. The authors seem to dismiss this possibility (e.g., lines 305 and 306), and favor a model where isotypes uniquely alter the propagation of allosteric effects across the lattice. But the finding that higher concentrations of taxol are required to alter β 3 microtubules could also be explained by weaker binding of β 3 to taxol. Either scenario could create a requirement for higher taxol concentrations in order to reach a threshold of microtubule stabilizing/expanding effects. Testing the

binding of taxol to microtubules assembled from different isotypes may be beyond the scope of this manuscript, but at minimum this alternative model should be considered and included in the discussion. Prior studies that test this possibility should be described and cited (e.g., PMID: 27651486).

2) The use of zebrafish and human $\beta 4$ heterodimers throughout the study is confusing. Figure 4 shows that these heterodimers are not functionally equivalent, yet other experiments in the study seem to use them interchangeably. This leads to confusion; for example, Figure 1e and f show that multi-phase depolymerization is a property of zebrafish $\beta 4$ tubulin. But it is not clear whether human $\beta 4$ heterodimers exhibit similar effects. The authors should use a consistent and homogeneous source of $\beta 4$ heterodimers throughout the study to make the results comparable across experiments. Comparing human $\beta 4$ heterodimers to human $\beta 3$ heterodimers would make the most sense.

Additionally, the use of tubulins from different species is potentially problematic for the segmented microtubule experiments because it is unknown whether zebrafish $\beta 4$ tubulin will assemble into the ends of microtubules made of human $\beta 3$ in a manner that is similar to how human $\beta 4$ would assemble. This raises the possibility that differences in the segmented microtubules could be explained by lattice defects induced by creating a microtubule from two different species of tubulin.

3) In several places, the study is lacking in statistical analyses that are necessary to support the authors' conclusions. Examples lacking statistical analysis include the following:

i) The growth rates in Figure 1g and h are described as similar, but the rates in Figure 2d and e are described as different, despite the respective values being very similar. It is unclear what statistical analyses are used to make these determinations.

ii) Figure 4f. Which gliding velocities are different?

iii) Figure 4h. What statistical analyses are used to compare these changes in gliding angle?

iv) Figure 5. The rates in these plots are described as having a non-linear relationship to the proportion of $\beta 3$ tubulin. What test was used to determine linearity? We could not find these details.

4) Figure 1. The results reported here should also include frequencies of catastrophe and rescue. Catastrophe is particularly important because this is the main difference reported in previous studies of isotypically pure tubulins by the Kapoor and Roll-Mecak labs. We suggest that the authors include a table of microtubule dynamics measurements that includes their results from different isotypes, different species, blends of isotypes, and comparing isotypes in different concentrations of taxol. This would enable the reader to easily compare the results across conditions. This table should also include appropriate statistical tests to support the main conclusions from pairwise comparisons.

5) Line 160-161. The authors should refrain from making statements about the intersection between $\beta 3$ and $\beta 4$ segments of microtubules in this experiment. Without differentially labeling the different isotypes, how can the authors identify the location of the intersection or whether the intersection is abrupt?

6) The analysis of kinked microtubules in Figure 3 is currently insufficient. It is not clear how kinks are defined and identified. For example, there appear to be a kinked microtubule in the two left panels of $\beta 3$ microtubules in Figure 3A. Would this be counted as a kinked microtubule? What is the relative abundance of kinks and how are kinks defined? A more quantitative (linearity or tortuosity) and less binary measurement might allow the authors to measure the relative straightness of microtubules made from different tubulins, and detect subtler distortions in the lattice that do not produce an obvious kink.

The authors interpret kinks as local lattice expansion. An important test of this model is whether this is indeed a local effect. This could be tested by using fluorescent taxol described by the Akhmanova lab to show that kinks occur at sites of high taxol binding. A second test would be whether higher taxol concentrations eliminate kinks by achieving a state where the lattice approaches uniform

expansion. Either test would strengthen the conclusion.

7) The writing, particularly in the results section, is sparse and difficult to follow at times. Many of the subsections of the results simply state the experimental results, without establishing the prediction and the rationale supporting it. This makes the logical flow of the study difficult to follow and gives the impression that the results are simply off the cuff observations, rather than tests of important hypotheses. Furthermore, the results text often lacks key details that are important for the reader to understand the design of the experiments. These concerns apply to the sections beginning on lines 180, 199, 228, 258 and 268.

On a related note, several of the figures could be re-ordered to improve the flow of the manuscript. Figure 5 a and b should be added into Figure 1 and included in the comprehensive dynamics table (point #4, above). Figure 5c could be added as a panel in Figure 4. Figure 6 a-c are not necessary for the reader to understand the manuscript, and could be moved to the supplement.

Minor concerns:

- 1) Figure 1 d-f. The description of the scalebars in the legend is incorrect; the horizontal and vertical lines are flipped.
- 2) Figure 2. The manuscript should include a methods paragraph clearly describing how the multi-segmented microtubules were created, in addition to the description in the results section.
- 3) Figure 3b and c. What are the arrowheads pointing to? This should be stated in the figure legend.
- 4) Figure 3d. The combination of green, pink, and red colors may be difficult to distinguish for people with color vision deficiency. The authors should consider a different color scheme. ASCB has a helpful article on this topic:
<https://www.ascb.org/science-news/how-to-make-scientific-figures-accessible-to-readers-with-color-blindness/>
- 5) Line 46. No need to capitalize the generic name paclitaxel.
- 6) Line 46. To our knowledge, colchicine was the first microtubule drug to be discovered.
- 7) Line 68. While it is true that single point mutants in isotypes have been linked to disease, previous studies in genetic models of isotype knockouts show that not all isotypes are essential for life. This is an important point that the authors should convey.
- 8) Line 200. "Kinesin can sense a taxol-dependent difference..." The authors could be more specific here; this has been shown for kinesin-1.
- 9) Does the manuscript include callouts to the Supplemental Figures 1 and 2? We could not find them.
- 10) Fig S2. The human tubulin sample in lane 1 shows smearing across higher molecular weights. Can the authors comment on the potential source(s) of this smearing? Is this tubulin or a contaminant?

Signed:
Jeff Moore
University of Colorado AMC

We are grateful to all 3 of our expert reviewers for their careful reading and constructive comments. We believe we have addressed all the comments and that doing so has improved the manuscript. We have marked the changes we have made with coloured text, and respond point-by-point below:

Reviewer #1 (Remarks to the Author):

In their paper, Chew and Cross use in vitro reconstitution with purified components to tackle multiple questions in the microtubule field about the functions of taxol, lattice expansion and compaction, and tubulin isotype differences. One of the most fundamental parts of this paper is that the authors are able to differentiate between the stabilizing and expanding functions of taxol using two tubulin isotypes: $\alpha1\beta4$ and $\alpha1\beta3$. They find that $\alpha1\beta4$ lattices are both stabilized and expanded by taxol, while $\alpha1\beta3$ lattices can only be stabilized by taxol (and only at much higher concentrations). They perform a functional assay with kinesin-1 and find that taxol accelerates kinesin gliding of $\alpha1\beta4$ lattices, but not $\alpha1\beta3$ lattices. This is consistent with this lab's prior work showing kinesin-1 can actively expand the GDP lattice, but further implies kinesin-1 will preferentially bind an already expanded lattice over a compacted lattice.

The study is well performed, the results are exciting, and the discussion is engrossing.

My only suggestions are 1) to include a reference to our recent work showing that tau can compact the lattice (Siahaan et al., 2022) within the first paragraph of the introduction when the authors list EB1 and doublecortin

> Done (ref 13)

and 2) to expand a bit in the discussion about other implications of their work. For example, do the authors hypothesize that $\alpha1\beta3$ lattices are resistant to expansion? I find this to be an intriguing possibility, but the authors have more knowledge on this topic than I. It could be that certain lattices are kept in either an expanded or compacted state due to MAPs that prefer one state over the other. OR, it could be that certain lattices are capable of being expanded and certain lattices are simply not.

Also, what do the authors think about the physiological purpose of these different isotypes within different cell types. The authors hint about $\beta3$ tubulin being highly abundant in neurons and that downregulating this isoform boosts kinesin-1 motility. Would you speculate that $\beta3$ lattices are preferred by other motors and that non- $\beta3$ lattices exist specifically for kinesin-1? Or, since kinesin-1 can obviously still associate with $\alpha1\beta3$ lattices, is this a mechanism to control the speed of kinesin-1-driven cargo in particular?

If the authors don't feel comfortable speculating, I understand (though I would still like to read their thoughts in response to this review). It's a great study that should be published regardless.

> We are pleased to be invited to speculate a little and have inserted the following as the second to last paragraph in our Discussion:

"What might be the biological significance of isotype-specific conformational switching of the microtubule GDP-lattice? We think that, since tubulin isotypes freely copolymerise, the interesting problem is the extent and mechanism by which each isotype influences the properties of the composite lattice, both locally and globally. We speculate that a tipping point might occur, whereby the conformation and properties of the lattice switch over in a concerted (avalanche) manner when a particular proportion of one or another isotype is reached. If this happens, then 'switchability' might

be favoured by a mix of isoforms that accesses the tipping point. Particular MAPS might bind preferentially to a particular isoform within the lattice, so that certain MAPS might load preferentially at particular isoform ratios. Conversely, particular MAPS or sets of MAPS might bias the recruitment of particular tubulin isoforms into the lattice. The biological value of this might lie in the potential for feedback regulation of microtubule-based processes created by the co-dependence of isoform ratio, the recruitment of particular MAPS, and the tendency of the lattice to converge to a particular conformation / expansion state.”

> Kinesin-1 in its rigor state can expand the lattice, but as we report here, lattice isoform influences the kinesin walking rate. Exactly how this works (especially under load) is something we are actively investigating, using optical trapping to interrogate single kinesin molecules walking under defined loads on different single and mixed isoform lattices.

Reviewed by Cassandra Ori-McKenney

Reviewer #2 (Remarks to the Author):

I read with interest the manuscript by Chew and Cross concerning the differential effect of Taxol on tubulin isotypes. I found the manuscript of interest, timely and I feel that it can be published in Communications Biology after revision.

My main concern is the fact that the interpretation of the observations relies on two hypothesis that does not seem to be supported by bibliography. First, individual tubulin subunits in the lattice can be either in the expanded or the compressed state, therefore lattices can be inhomogeneous. And second, $\beta 3$ tubulin subunits do not expand upon Taxol binding. Authors should consider the evidences supported by others references to build up a more suitable hypothesis considering their results.

They mention Ref. 35 and 18 in their manuscript. Ref 35. (which is now eLife 12:e84791) studies the effect of Taxol, docetaxel and other taxanes on microtubule lattice using fiber diffraction technique and purified calf brain tubulin.

Fiber diffraction measurements rely on diffraction selection rules for helical structures, which depend on helical samples with a homogenous pitch. When several layer lines are observed it is because the microtubules have a homogeneous lattice. In these experiments authors used calf-brain tubulin that roughly contains 33% of $\beta 3$, 50% of $\beta 2$ and likely about 10% of $\beta 4$. Their images show a homogeneous expansion of the microtubule lattice upon occupancy of the taxane site indicating that $\beta 3$ tubulin isotype is able to expand.

And this is not the only evidence because similar results are also shown in reference 18. Here, microtubules are assembled from porcine brain tubulin (similar isotype content) and, are also expanded when bound to Taxol.

> Thank you for these insightful comments. We did not mean to suggest that a mixed-isotype lattice can be conformationally inhomogeneous, except in the special case where segments of the same microtubule are built from different single isotypes, and these segments have different conformations (as evidenced by different segments gliding at different rates, Fig.4b-e). We have carefully been through our entire text and made minor revisions (changes marked with coloured text throughout) to avoid any impression to the contrary. Our data demonstrate that pure single isotype $\beta 3$ lattices do not expand in taxol (as reported by their slow motility in kinesin motility assays), but we of course agree that $\beta 3$ tubulin subunits can be expanded by taxol in the context of other isotypes (such as $\beta 4$) that bind taxol more tightly. We appreciate and indeed we already reference the Kamimura fibre diffraction experiments on the action of taxol on the lattice spacing of brain microtubules. Supplementary figure S3 shows that brain microtubules are stabilised by 200 nM taxol without much accelerating their gliding velocity, which we take to indicate they are compacted. Gliding of these same microtubules is accelerated by 10 μ M taxol, consistent with expansion of the $\beta 3$ component within a mixed-isotype lattice, as implied by the Kamimura data. We did not previously discuss these points relating to mixed isotype brain tubulin in our original ms but now, encouraged by the reviewer's comments, we do so:

(from para 4 of Discussion):

".. Exactly how switching between lattice states is driven by taxol binding to individual heterodimers within the lattice remains to be seen. In mixed isotype brain microtubules, maximum expansion requires stoichiometric amounts of taxol²². Our finding that substoichiometric taxol can induce kinks in single isotype $\alpha 1\beta 4$ microtubules suggests that taxol can induce local expansion of an otherwise compacted single isotype lattice. Our working model does not address the mechanism(s) by which the

lattice can be driven to change conformation – it seeks only to encapsulate a minimal hypothesis in which the GDP-tubulin lattice can be shifted between exactly 2 different effector-stabilised conformations, even though known stabilising effectors (GMPCPP, taxol and other effectors, for example rigor kinesin¹¹) interact at widely separated sites on β tubulin. We note that ‘Expanded’ and ‘compacted’ are catch-all descriptors for complex conformational changes that include lateral as well as axial components. It is possible that more than two effector-stabilised states of the GDP-tubulin lattice exist; but we only need two to explain our data.”

A final observation from reference 18, which is opposite to previous hypothesis, stated that microtubules require to be fully bound to Taxol to expand because in the presence of taxol:tubulin ratios lower than 1, as it is in this manuscript, microtubules keep a compressed lattice. Of course, this bulk solvent technique does not allow to identify local expansions.

> We find that with mixed isotype brain microtubules, 200 nM taxol is sufficient to stabilise (consistent with much earlier work (Derry et al , 1995)). By contrast, 10 μ M taxol (consistent with stoichiometric binding) is required to accelerate/expand. This in turn is consistent with Kamimura’s titration of taxol-induced lattice expansion of brain microtubules using x-ray fibre diffraction. We now note this point:

(Discussion para 6, p14)

“.. β 3 tubulin within mixed isotype microtubules binds less taxol than other isotypes in the lattice³⁶. Further, it is clear that taxol in sufficient amounts can expand mixed isotype lattices that include β 3 tubulins¹⁸. Consistent with these findings, we find that 10 μ M taxol accelerates the kinesin-driven gliding of mixed isotype brain microtubules, indicating lattice expansion, whilst 200-400 nM taxol stabilises but only accelerates a subpopulation, indicating lattice compaction in the slow subpopulation (Supp. Fig. S3). Within single isotype lattices, we hypothesise that α 1 β 3 tubulin has a lower taxol affinity than α 1 β 4 tubulin, and that the α 1 β 3 lattice cannot achieve the higher taxol occupancy required for expansion.”

Finally, data from Estevez Gallego et al (2020) 10.7554/eLife.50155 (not referenced in this manuscript and with experiments done in a similar way than ref. 18 and 35), shows that Taxol induced lattice expansion can be reverted by the occupancy of the β -phosphate site with phosphate analogs.

> We now reference the Estevez Gallego et al work when discussing GMPCPP-induced expansion (p11, ref 43). Reviewers 1 and 2 both felt we needed to reference more prior work. The journal format is a factor; but we have nonetheless now taken care in our Introduction to refer to key work on effector-driven expansion in general and taxol-induced expansion in particular. The ref list has now gone to 56, we hope this is acceptable.

Taxol to tubulin ratios employed in this work are far below those employed in the references mentioned above and, therefore local expansion in certain parts of the microtubule (kinks) might be possible. Also, it is possible that neighbor subunits have a strong influence in the conformation of a certain tubulin dimer.

> We agree with both these points

Considering their results and literature evidences it seems more plausible that the differential effect of Taxol on β 3 and β 4 tubulin isotypes is related to differences on binding affinities of the drug to these. Of course, it might also be possible that β 3 subunits can only expand when in inhomogeneous lattice due to the effect of neighbor subunits. However, in the absence of other structural data I find

the claim that $\beta 3$ tubulin subunits cannot expand is and overestimation of their results and, contradictory with previous structural studies.

> We in no way meant to claim that $\beta 3$ subunits cannot expand. To the contrary, we demonstrate that in AMPPNP, $\beta 3$ and $\beta 4$ subunits expand equivalently, as judged by the acceleration of kinesin-driven MT gliding. **Supp. Fig S3** shows expansion-acceleration of mixed isotype brain MTs, consistent with the earlier work the reviewer refers to and implying that $\beta 3$ subunits can be driven by taxol to expand when they represent a minority lattice component. Likewise, we did not mean to appear to reject the possibility that the differential expansion of $\beta 3$ and $\beta 4$ in taxol is due to their having different affinities for taxol – indeed we think this is the most likely explanation. We now state this in our Abstract:

“..In GMPCPP, single isotype $\alpha 1\beta 3$ and $\alpha 1\beta 4$ lattices both show accelerated gliding, establishing that both can in principle be driven to expand. Accordingly, we propose that taxol-induced lattice expansion requires a higher taxol occupancy than taxol-induced stabilisation, and that single isotype $\alpha 1\beta 3$ lattices are restricted to low taxol occupancy.”

In summary, I find that, in order to the manuscript be accepted, the authors should first, soften the specific claim that $\beta 3$ tubulin subunits cannot expand. Second, they should consider that Taxol can have different affinities for $\beta 3$ and $\beta 4$ subunits, which could in turn produce other lattice conformational changes different to expansion (e.g. lateral angle between protofilaments that could also affect kinesin motility). And third, they should discuss the apparent contradiction with previous structural results.

> See immediately above, we have revised the Abstract to state that GMPCPP can expand both $\beta 4$ and $\beta 3$, demonstrating that $\beta 3$ can in principle expand.

> See immediately above, we have revised the Abstract to include an explicit declaration that the differences we observe between $\beta 3$ and $\beta 4$ likely relate to taxol occupancy ($\beta 3$ can expand in principle but cannot achieve a high taxol occupancy).

> See above, we believe our results do not contradict previous work and we now discuss this.

Minor concerns.

Why does the authors select $\beta 4$ tubulin and not the more common $\beta 1$ and $\beta 2$ isotypes?

> We chose $\beta 3$ on the grounds it is largely confined to neurons, and overexpressed in taxol-resistance. We chose $\beta 4$ over $\beta 2$ because the two are very similar and $\beta 2$ is well studied already. So we tried, as ever, to make the most progress we could with limited resources. We now explain our rationale in more detail in the introductory paragraph of our Results section.

I wonder why the authors co-assembly zebrafish $\beta 4$ tubulin with human $\beta 3$ tubulin instead of sticking to human proteins (especially considering the differences between both $\beta 4$ isotypes found, Figure 4a)

> See immediately above, we studied zebrafish $\beta 4$ because we wanted to begin to validate the zebrafish system as a model for human tubulinopathies. Given that our interest is in the potential of taxol to have different actions on different isotypes, there is no reason in principle to limit ourselves to human isotypes – taxol is after all, not just used on humans. Zebrafish $\beta 4$ and human $\beta 4$ are extremely similar in sequence (Fig. 6a). We expected to be able to show an exact equivalence of human $\beta 4$ and zebrafish $\beta 4$. In the event we found very slight differences, so we report these. We

now explain these points in the introductory paragraph of our Results section, and have added a short paragraph to the Discussion about the properties of Hs β 4 versus Dr β 4:

(p15) “ We worked predominantly with Dr β 4 tubulin, but we included Hs β 4 in some experiments. Dr and Hs β 4 differ at only 5 residue positions, all surface exposed. 3 are in the regions that engage the M-loop of the lateral nearest neighbour and the other 2 are at the C-terminus (Fig. 6a). We expected Hs β 4 and Dr β 4 tubulins to behave alike, and they do have similar properties (Fig.1h-l). Remarkably however, we found that at 400 nM taxol, Dr β 4 lattices appear predominantly expanded, yet Hs β 4 appear predominantly compacted, as judged by their gliding velocities (Fig. 4a). Our evidence thus suggests that a very few, apparently conservative residue substitutions may detectably shift the taxol-dependence of β 4 lattice switching. ”

Note that the comparison between GMPCPP stabilized microtubules and Taxol stabilized microtubules is not rigorous because in the conditions of the study GMPCPP stabilized microtubules are homogeneous (they have 100% GMPCPP bound subunits) while taxol stabilized microtubules have only a percentage of tubulin molecules bound.

> Our goal here was to confirm that β 3 single isotype MTs can in principle expand their spacing, given a suitable allosteric effector. GMPCPP of course works differently to taxol – it binds to a different site and directly manipulates a different part of the tubulin structure. As noted above, we have edited the Abstract to clarify.

Note that Taxol bound and unbound microtubules are not only different in the axial direction but also in the lateral direction (different number of protofilaments) Ref 35 and 10.1016/j.str.2022.11.006 and that this change is extremely easy and quick 10.1074/jbc.273.50.33803 this should be considered as well when discussing the slow and fast gliding lattices.

> Thank you for this comment. We agree and we now explicitly state that ‘expansion’ and ‘compaction’ are catch-all descriptors that necessarily simplify a complex (potentially very much more complex) picture:

(Discussion, para3)

“We note that ‘Expanded’ and ‘Compacted’ are catch-all descriptors for complex conformational changes that include lateral as well as axial components. It is possible that more than two effector-stabilised states of the GDP-tubulin lattice exist; but we only need two to explain our data.”

Line 40. Reference 2 ins not related to this sentence. > Corrected

Line 41. Consider citing Estevez Gallego et al (2020) 10.7554/eLife.50155 for the effect of γ -phosphate analogs and GMPCPP vs GMPPCP. > Added as ref 43.

Line 46. Colchicine was the first tubulin directed drug to be discovered, and Vinca alkaloids the first in clinical use. > We have corrected the text to read “ .. first microtubule stabilising drug ..”

Lines 48-51. Consider to reference these sentences. > We have included as many references as we reasonably can given the journal guideline is 50.

Line 54. Ref 35 describe crystallographically precise molecular mechanisms of microtubule stabilization of taxanes.

> Ref 35 indeed establishes precise structural details of taxol binding, whilst also confirming that taxol binds much more tightly (4 orders) to the lattice than to free tubulin. Discovering how the action of taxol varies by isotype for lattices of different tubulin isotypes is a helpful way to test and validate mechanistic ideas. We have adjusted our wording, which we accept was inexact.

“ Crystallographically precise details of the taxol binding site are now established²², but the molecular mechanisms by which taxol shifts the conformation of lattices of polymerised tubulin heterodimers are less clear. “

Line 63. Ref 20 does not show that Taxol can assemble GDP-tubulin molecules, only that it does not require external GTP. Consider citing Díaz and Andreu Biochemistry 32, 2747-2755 (1993) Assembly of Purified GDP-Tubulin into Microtubules induced by Taxol and Taxotere: Reversibility, Ligand Stoichiometry, and Competition.

> We agree and now cite both refs (refs 25 & 26)

Line 128. From d. Should be from e > Fig leg Corrected

Line 131: μ : mean; I miss the units of μ > Fig leg corrected

Line 230: There is no figure 4i! > Corrected

Lines 385-387: References 39 and 40 are the same. > Corrected

Line 593: Hope the channel volume is 10 μ l not 10 ml > Corrected

Line 597: The same as 593 > Corrected

Line 664: 40 mM tubulin is 4.4 g/ml seems a bit too much! > Corrected

Line 677: 10 mM tubulin is 1 g/ml in general PLEASE CHECK THE UNITS ALL ALONG THE MANUSCRIPT > m for μ corrected throughout

Reviewer #3 (Remarks to the Author):

This study by Chew et al examines whether tubulin isotypes exhibit different rates of microtubule dynamics and sensitivity to the drug, taxol. These are important questions in the microtubule field. We have known about tubulin isotypes for several decades, but whether isotypes exhibit distinct activities and drug sensitivity has remained incompletely understood. This question is particularly important for the chemotherapeutic drug, taxol. Taxol resistance is associated with aberrant isotype expression, but we do not understand how isotypes determine taxol sensitivity.

To address these questions, the authors use recombinant approaches to generate isotypically pure samples of tubulin. They use these samples to show that heterodimers containing $\beta 3$ tubulin exhibit faster depolymerization than heterodimers containing $\beta 4$ tubulin. Furthermore, $\beta 3$ heterodimers maintain microtubule dynamics at concentrations of taxol that dampen the dynamics of $\beta 4$ tubulin. The authors provide evidence that taxol generates a lattice state in microtubules assembled from $\beta 4$ tubulin that supports faster rates of kinesin motility; however, taxol fails to generate a similar lattice state in microtubules assembled from $\beta 3$ tubulin. This is an important finding that establishes a new mechanistic explanation for how $\beta 3$ expression levels may determine cellular response to taxol treatment. The authors also show that this effect exhibits a non-linear relationship to the concentration of $\beta 3$ tubulin in the microtubule, suggesting that $\beta 3$ may influence the structure of neighboring tubulins through an allosteric mechanism.

The study addresses important questions using powerful and recently developed techniques. While some of the results confirm published findings by other labs, the results here go further toward a unified view of how isotype composition might propagate distinct states of the lattice that could influence microtubule dynamics, kinesin activity, and drug response. While we are enthusiastic about the study, there are weaknesses in the design, interpretation and writing of the current manuscript that must be amended prior to publication. Addressing these concerns would improve clarity and strengthen the conclusions of the study.

Major concerns:

1) A central weakness of the study is that the authors do not measure taxol binding to microtubules containing $\beta 3$ vs $\beta 4$ heterodimers. This is an important point because the apparent insensitivity of $\beta 3$ tubulin to levels of taxol that are sufficient to strongly affect $\beta 4$ could be due to isotype-specific differences in binding affinity. The authors seem to dismiss this possibility (e.g., lines 305 and 306), and favor a model where isotypes uniquely alter the propagation of allosteric effects across the lattice.

> As noted above, we did not intend to appear to dismiss this possibility, indeed we think this is the most likely explanation for the differences between $\beta 3$ and $\beta 4$ single isotype lattices. We now explicitly state in our Abstract:

“ In GMPCPP, single isotype $\alpha 1\beta 3$ and $\alpha 1\beta 4$ lattices both show accelerated gliding, indicating that both can in principle be driven to expand. Accordingly, we propose that taxol-induced lattice expansion requires a higher taxol occupancy than taxol-induced stabilisation, and that single isotype $\alpha 1\beta 3$ lattices are restricted to low taxol occupancy.”

But the finding that higher concentrations of taxol are required to alter $\beta 3$ microtubules could also be explained by weaker binding of $\beta 3$ to taxol. Either scenario could create a requirement for higher taxol concentrations in order to reach a threshold of microtubule stabilizing/expanding effects.

Testing the binding of taxol to microtubules assembled from different isotypes may be beyond the scope of this manuscript, but at minimum this alternative model should be considered and included in the discussion.

> As requested, we now explicitly discuss these points, including the possibility that a taxol-dependent conformational change in the subunit that binds taxol might shift the conformation of its neighbours, potentially enhancing their affinity for taxol. As we discuss, a taxol-driven allosteric shift that partially propagates to neighbouring subunits in the lattice might begin to explain how taxol binding generates the localised zones of lattice expansion (kinks) that we observe in beta 4 lattices.

(Discussion para 6)

".. taxol induces kinks in Dr $\alpha 1\beta 4$ but not Hs $\alpha 1\beta 3$ microtubules and that taxol washout relaxes the kinks (Fig. 3), consistent with them being due to reversible, taxol-dependent local expansion of the $\alpha 1\beta 4$ single isotype GDP-tubulin lattice, possibly at defects."

(Discussion para 7)

"How does taxol stabilise and expand single isotype $\alpha 1\beta 4$ lattices, yet stabilise single isotype $\alpha 1\beta 3$ lattices without expanding them? The simplest hypothesis is that lattice expansion requires a higher taxol occupancy than lattice stabilisation (Fig. 6d). There is robust evidence that $\beta 3$ tubulins within mixed isotype microtubules bind less taxol than other isotypes in the lattice³⁶. Nonetheless it is clear that taxol in sufficient amounts can expand mixed isotype lattices that include $\beta 3$ tubulins¹⁸, suggesting that $\beta 3$ tubulins within a mosaic isotype lattice can be driven to expand via taxol binding to their neighbours. In this way the affinity of $\beta 3$ tubulin for taxol might be context-dependent, potentially leading to positive feedback effects in which local lattice expansion, induced by taxol, promotes further taxol binding and further expansion. By contrast in single isotype lattices, we hypothesise that $\alpha 1\beta 3$ tubulins have a much lower taxol affinity than $\alpha 1\beta 4$ tubulins, and so cannot achieve the higher taxol occupancy required for expansion."

Prior studies that test this possibility should be described and cited (e.g., PMID: 27651486).

> We previously cited an excellent review of this whole area, including the 2-(m-Azidobenzoyl)taxol results, by these same authors (ref 17). We still do so, but we now as requested also directly reference the study referred to, of the binding of 2-(m-Azidobenzoyl)taxol binding to specific tubulin isotypes in different lattices (ref 36).

2) The use of zebrafish and human $\beta 4$ heterodimers throughout the study is confusing. Figure 4 shows that these heterodimers are not functionally equivalent, yet other experiments in the study seem to use them interchangeably. This leads to confusion; for example, Figure 1e and f show that multi-phase depolymerization is a property of zebrafish $\beta 4$ tubulin. But it is not clear whether human $\beta 4$ heterodimers exhibit similar effects. The authors should use a consistent and homogeneous source of $\beta 4$ heterodimers throughout the study to make the results comparable across experiments. Comparing human $\beta 4$ heterodimers to human $\beta 3$ heterodimers would make the most sense.

Additionally, the use of tubulins from different species is potentially problematic for the segmented microtubule experiments because it is unknown whether zebrafish $\beta 4$ tubulin will assemble into the ends of microtubules made of human $\beta 3$ in a manner that is similar to how human $\beta 4$ would assemble. This raises the possibility that differences in the segmented microtubules could be explained by lattice defects induced by creating a microtubule from two different species of tubulin.

> We explain our rationale for working with two different $\beta 4$ isotypes in the first paragraph of our Results section.

".. We studied both Homo sapiens (Hs) and zebrafish (Danio rerio, Dr) $\beta 4b$ isotypes to test whether their very minor sequence differences influence function. Tubulin mutants in this model organism can potentially provide insight into human tubulinopathies^{29,32}."

> It is certainly possible that defects tend to occur between the segments in segmented isotype microtubules. We see no reason to think this risk or property would necessarily be absent for human-human segmented microtubules. Finding out will require electron microscopy and we will do this, but this lies out of scope for the present report.

3) In several places, the study is lacking in statistical analyses that are necessary to support the authors' conclusions. Examples lacking statistical analysis include the following:

i) The growth rates in Figure 1g and h are described as similar, but the rates in Figure 2d and e are described as different, despite the respective values being very similar. It is unclear what statistical analyses are used to make these determinations.

> In Fig 1g,h, the growth rates are both similar (16.4 versus 19.2 nm/sec) and statistically significantly different (**, Fig. 2d).

ii) Figure 4f. Which gliding velocities are different?

> Thank you for this comment. Significance brackets have been added.

iii) Figure 4h. What statistical analyses are used to compare these changes in gliding angle?

> Prompted by the reviewer's question, we applied a 2-sample Kolmogorov-Smirnov test, to quantifies the probability that two cumulative density distributions are different from one another. The p-values confirm that the changes in direction accumulate at rates that indeed depend on isotype composition, as we suggested. Significance brackets added to Fig. 4j.

iv) Figure 5. The rates in these plots are described as having a non-linear relationship to the proportion of $\beta 3$ tubulin. What test was used to determine linearity? We could not find these details.

> We are uncomfortable trying to fit a definite sigmoidal function, because we don't have a clear enough idea of the nature of the apparent cooperativity. To try to address this comment we have fitted a linear function. In the cases we refer to (depolymerisation rates and gliding rates versus isotype ratio), the poor fit is obvious by eye. Note that these plots display all the raw data, ensuring the reader gains a faithful impression of their distribution.

4) Figure 1. The results reported here should also include frequencies of catastrophe and rescue. Catastrophe is particularly important because this is the main difference reported in previous studies of isotypically pure tubulins by the Kapoor and Roll-Mecak labs. We suggest that the authors include a table of microtubule dynamics measurements that includes their results from different isotypes, different species, blends of isotypes, and comparing isotypes in different concentrations of taxol. This would enable the reader to easily compare the results across conditions. This table should also include appropriate statistical tests to support the main conclusions from pairwise comparisons.

> We already reference relevant work from the Kapoor & Roll-Mecak labs. Encouraged by the reviewer, we now provide numbers for the dynamic instability parameters of pure $\alpha 1\beta 3$ and $\alpha 1\beta 4$ lattices in the absence of taxol (Fig 1l). In the presence of taxol, catastrophe does not happen, so neither does rescue. For consistency with previous work, we fit normal distributions and provide means and sd, even though, as we show in Fig.1i,j, the depolymerisation data are clearly not normally distributed.

5) Line 160-161. The authors should refrain from making statements about the intersection between

$\beta 3$ and $\beta 4$ segments of microtubules in this experiment. Without differentially labeling the different isotopes, how can the authors identify the location of the intersection or whether the intersection is abrupt?

> We built the segmented isotype lattices for dynamics experiments by first growing one single isotype lattice and then washing out that tubulin and switching to the other. The switchover between single isotype heterodimers is fast and we can inspect the microtubules before and after washout, so we are rather sure where the intersection lies. We have added a new para to the Methods section to explain.

6) The analysis of kinked microtubules in Figure 3 is currently insufficient. It is not clear how kinks are defined and identified. For example, there appear to be a kinked microtubule in the two left panels of $\beta 3$ microtubules in Figure 3A. Would this be counted as a kinked microtubule? What is the relative abundance of kinks and how are kinks defined? A more quantitative (linearity or tortuosity) and less binary measurement might allow the authors to measure the relative straightness of microtubules made from different tubulins, and detect subtler distortions in the lattice that do not produce an obvious kink.

> We agree in principle but tracking and quantifying the kinking as the microtubules wave around in solution is unfeasible in 3D dark field because the large depth of field equates to poor Z-axis resolution so that we cannot see kinks in the Z direction. The X-Y components of kinks, or their absence, are most obvious in the movies (Supp. mov. M1-3), because the thermal waving around of the microtubule guides the eye to kinks, whose axial position remains fixed as the microtubule grows. We are gearing up to study this phenomenon more carefully using electron microscopy, which will allow us to visualise the fine-structure of kinks properly and, we hope, detect otherwise-invisible smaller defects and kinks. We hope the reviewer will allow that for the moment, the reversible induction by taxol of obvious (larger) kinks, visible by eye, is worth reporting.

The authors interpret kinks as local lattice expansion. An important test of this model is whether this is indeed a local effect. This could be tested by using fluorescent taxol described by the Akhmanova lab to show that kinks occur at sites of high taxol binding. A second test would be whether higher taxol concentrations eliminate kinks by achieving a state where the lattice approaches uniform expansion. Either test would strengthen the conclusion.

> It would indeed be great to visualise taxol binding, but our preliminary tries with various fluorescent taxols did not give us confidence that they behave like 'real' taxol. They bind more weakly, and they are bad at stabilising and potentially at expanding the lattice. We now mention this in the text:

(Discussion, para 7)

".. in our hands fluorescent taxols stabilise the GDP-lattice only marginally, which limits their value as reporters. "

> We will continue to explore this point.

> The Akhmanova lab work referred to identified preferential binding of fluorescent taxol at defects and we reference this already (ref 28).

7) The writing, particularly in the results section, is sparse and difficult to follow at times. Many of the subsections of the results simply state the experimental results, without establishing the prediction and the rationale supporting it. This makes the logical flow of the study difficult to follow

and gives the impression that the results are simply off the cuff observations, rather than tests of important hypotheses. Furthermore, the results text often lacks key details that are important for the reader to understand the design of the experiments. These concerns apply to the sections beginning on lines 180, 199, 228, 258 and 268.

> This is a difficult comment to address! We indeed, necessarily, sought brevity - even as it stands the paper is full-length for the journal guidelines. We have been back through the paper and tried to ensure we have clearly explained the question(s) that each experiment addresses. In most cases the hypothesis is simply the null hypothesis, that there is no difference between the isotypes in their response to taxol, and we set out to disprove that. Each point where we have added additional explanation is marked with coloured text. In response to comments from this and our other reviewers, we have revised the abstract to make room for a more complete description of the model we are proposing, based on our results.

On a related note, several of the figures could be re-ordered to improve the flow of the manuscript. Figure 5 a and b should be added into Figure 1 and included in the comprehensive dynamics table (point #4, above). Figure 5c could be added as a panel in Figure 4. Figure 6 a-c are not necessary for the reader to understand the manuscript, and could be moved to the supplement.

> Fig. 5 is about the apparently nonlinear dependence of depolymerisation rate and kinesin-driven gliding velocity on the isotype ratio of mosaic isotype microtubules. Fig.1 is about the dynamic instability of pure single isotype lattices. We are reluctant to further expand the scope of Fig.1, we continue to think it makes sense first, via Fig.1, to present the properties of pure single isotype microtubules and then move on, via subsequent figures, to report the properties of mosaic isotype and segmented isotype lattices, with and without taxol. We have added a table of dynamic instability parameters to Fig. 1, as requested (Fig. 1I).

> Fig.4 is about the effect of taxol on gliding velocities, Fig. 5c fixes the taxol concentration and varies the tubulin isotype ratio. We feel the nonlinear dependence of gliding velocity and depolymerisation rate on isotype ratio is a novel and central finding – previous work on mosaic isotype lattices has not picked this up. We feel this point warrants its own figure and that trying to make this central point by referring to a mix of panels from other figures would be awkward.

> Figure 6a-c show the clustering of residue differences between $\alpha1\beta3$ and $\alpha1\beta4$ in the M-loop 'receptor' region on the nearest neighbour in the lattice. In all the previous work on the taxol response of $\alpha1\beta3$ this has gone unnoticed – attention has previously focussed entirely on differences in the M-loop region itself. We feel we need these panels in the main text to make this key point.

Minor concerns:

1) Figure 1 d-f. The description of the scalebars in the legend is incorrect; the horizontal and vertical lines are flipped. > Thank you, corrected

2) Figure 2. The manuscript should include a methods paragraph clearly describing how the multi-segmented microtubules were created, in addition to the description in the results section.
> Thank you for picking this up, we now do so.

3) Figure 3b and c. What are the arrowheads pointing to? This should be stated in the figure legend.
> They mark the kinks. We have corrected the figure legend.

4) Figure 3d. The combination of green, pink, and red colors may be difficult to distinguish for people

with color vision deficiency. The authors should consider a different color scheme. ASCB has a helpful article on this topic:

<https://www.ascb.org/science-news/how-to-make-scientific-figures-accessible-to-readers-with-color-blindness/>

> Thank you for spotting this. We have changed the red to black, and also been through the rest of the figures to check and correct this issue.

5) Line 46. No need to capitalize the generic name paclitaxel.

> Fixed

6) Line 46. To our knowledge, colchicine was the first microtubule drug to be discovered.

> Thank you for spotting this mistake, we have corrected this to 'first microtubule-stabilising drug to be discovered'.

7) Line 68. While it is true that single point mutants in isotypes have been linked to disease, previous studies in genetic models of isotype knockouts show that not all isotypes are essential for life. This is an important point that the authors should convey.

> Since we submitted this work, a thoughtful review of the complex area of mutation-specific disease mechanisms has appeared and we now reference this for a careful discussion of the biological role of tubulin isotypes, the ability of certain isotypes to partially substitute for others (giving rise to non-essentiality) and the cell biological impact of tubulin mutations, including those acting indirectly via MAP and motor function.

<https://www.frontiersin.org/articles/10.3389/fcell.2023.1136699/full>.

8) Line 200. "Kinesin can sense a taxol-dependent difference..." The authors could be more specific here; this has been shown for kinesin-1.

> Corrected to 'kinesin-1 (KHC)'.

9) Does the manuscript include callouts to the Supplemental Figures 1 and 2? We could not find them.

> Callouts added to Methods section.

10) Fig S2. The human tubulin sample in lane 1 shows smearing across higher molecular weights. Can the authors comment on the potential source(s) of this smearing? Is this tubulin or a contaminant?

> The smearing is due to salts in the sample. We have added an extra panel as **Supp Fig. S2b**, illustrating the serial tag affinity purification steps for this isotype, including a lane of the purified protein, without visible contamination and with no smearing.

Signed:

Jeff Moore

University of Colorado AMC

REVIEWERS' COMMENTS:

Reviewer #1 (Remarks to the Author):

The authors have addressed all of my comments. I look forward to seeing this paper in print and the subsequent studies on kinesin-1!

Reviewer #2 (Remarks to the Author):

The authors have adequately address my concerns, therefore the manuscript can be published as it is.

However, I would like to point towards the comment raised by Jeff Moore 3: The authors interpret kinks as local lattice expansion. An important test of this model is whether this is indeed a local effect. This could be tested by using fluorescent taxol described by the Akhmanova lab to show that kinks occur at sites of high taxol binding.

I agree that this might be a very interesting experiment that may enhance the quality of the manuscript.

The authors replied: It would indeed be great to visualise taxol binding, but our preliminary tries with various fluorescent taxols did not give us confidence that they behave like 'real' taxol. They bind more weakly, and they are bad at stabilising and potentially at expanding the lattice.

I agree that commercially available fluorescent taxanes bind weakly and are bad stabilizing. However, this is not the case for all fluorescent compounds, there are fluorescent compounds designed to have high affinity as F-Chitax3 which was succesfully used by Akhmanova's lab in the work mentioned.

Reviewer #3 (Remarks to the Author):

The authors have sufficiently addressed our concerns. The revised manuscript is clear and the additions to the discussion section do an excellent job of putting the study into context and identifying important next steps. This is an important contribution to the field and we look forward to seeing it in published form.

We found two typos the authors might address before final publication:

Abstract, line 14: Add 'of' so the sentence reads "...drives gliding of segmented-isotype GDP-taxol microtubules along..."

Figure 3A-C. Change the inset labels to read: "Hs $\alpha 1\beta 3$ " and "Dr $\alpha 1\beta 4$ "